# Cas9-assisted biological containment of a genetically engineered human commensal bacterium and genetic elements

Naoki Hayashi [1,2,10], Yong Lai[3,4,5,10], Jay Fuerte-Stone [6], Mark Mimee [6,7] & Timothy K. Lu[1,3,4,8,9]

Sophisticated gene circuits built by synthetic biology can enable bacteria to sense their environment and respond predictably. Engineered biosensing bacteria outfitted with such circuits can potentially probe the human gut microbiome to prevent, diagnose, or treat disease. To provide robust bio-containment for engineered bacteria, we devised a Cas9-assisted auxotrophic biocontainment system combining thymidine auxotrophy, an Engineered Riboregulator (ER) for controlled gene expression, and a CRISPR Device (CD). The CD prevents the engineered bacteria from acquiring *thyA* via horizontal gene transfer, which would disrupt the biocontainment system, and inhibits the spread of genetic elements by killing bacteria harboring the gene cassette. This system tunably controlled gene expression in the human gut commensal bacterium *Bacteroides thetaiotaomicron*, prevented escape from thymidine auxotrophy, and blocked transgene dissemination. These capabilities were validated in vitro and in vivo. This biocontainment system exemplifies a powerful strategy for bringing genetically engineered microorganisms safely into biomedicine.

The microbiome has recently become a point of focus for biomedical research. Altered microbial communities have been associated with a multitude of diseases, including inflammatory bowel diseases[1,2], liver disease[3], metabolic disorders[4], and cancer[5], as well as responses to COVID-19[6] and other infections[7]. Synthetic biology offers powerful tools for manipulating the microbiome and elucidating the functions of its various components. Genetically engineered bacteria and sophisticated genetic circuits provide the means to investigate the exchange of biological signals between human and microbial cells and to eventually gain control over processes that can shield the body from diseases or overcome diseases at their onset.

Stringent regulations limit the use of genetically engineered bacteria to prevent them from being unintentionally released into the environment. The potential benefit of these organisms can be realized only once we have found ways to comply with these regulations. Thus, before the tools of synthetic biology can be applied, for example, to produce a useful protein in situ[8], to stimulate immunity in response to cancer[9], or to destroy a harmful virus in a host[10], technologies are needed to contain the genetically modified organisms and the genetic material they carry, which could find its way into other cells and alter the microbiome in unpredictable ways.

[1]Department of Biological Engineering, Massachusetts Institute of Technology, Cambridge, MA 02139, USA. [2]JSR-Keio University Medical and Chemical Innovation Center (JKiC), JSR Corp., 35 Shinanomachi, Shinjuku, Tokyo 160-8582, Japan. [3]Synthetic Biology Group, MIT Synthetic Biology Center, Massachusetts Institute of Technology (MIT), Cambridge, MA 02139, USA. [4]Research Laboratory of Electronics, MIT, Cambridge, MA 02139, USA. [5]Department of Chemical and Biological Engineering, The Hong Kong University of Science and Technology, Clear Water Bay, Kowloon, Hong Kong SAR. [6]Department of Microbiology, The University of Chicago, Chicago, IL 60637, USA. [7]Pritzker School of Molecular Engineering, The University of Chicago, Chicago, IL 60637, USA. [8]Broad Institute, Cambridge, MA 02139, USA. [9]Harvard-MIT Division of Health Sciences and Technology, Cambridge, MA 02139, USA. [10]These authors contributed equally: Naoki Hayashi, Yong Lai. ✉e-mail: mmimee@uchicago.edu; tim@lugroup.org

Many forms of biological containment have been devised for uncontrolled environments that do not require human monitoring or input[11,12]. There are four main strategies for accomplishing biological containment: auxotrophy, kill switches, genetic firewalls, and physical containment. Auxotrophy takes advantage of engineered auxotrophic organisms, which are unable to synthesize an essential compound or to utilize an environmentally available compound required for their survival. The lack of the integral compound induces lethality outside the uncontrolled area[13]. The kill switch employs the conditional production of a toxic molecule whose gene expression is tightly controlled by an environmentally responsive element. Once engineered bacteria with the kill switch come into an undesirable environment, the switch is turned on and the bacteria are killed by the toxic molecule[14]. Genetic firewalls use alternative genetic codes in bacteria for nonstandard amino acids (NSAAs), enabling genetic isolation by preventing horizontal gene transfer (HGT). The growth of the bacteria harboring the genetic firewall thus depends on the use of the NSAAs, such that the firewall sets up robust and orthogonal barriers between the engineered organisms and the environment[15,16]. The artificial genetic code, incorporating NSAAs, makes the standard amino acids from natural microorganisms nonfunctional in the engineered bacteria. Additionally, as bacteriophage infection can lead to the HGT of bacterial genes, a genetic firewall has been created by the deletion of cellular transfer RNAs and release factor 1[17]. Physical containment involves trapping the engineered bacteria in a space with well-designed materials, such as a multilayer hydrogel capsule. The bacteria cannot escape but still have access to nutrients and chemical inputs from their outer environment[18]. These four strategies prevent the engineered strains and the genetic elements from escaping from controlled environments.

Auxotrophy, one of the most promising approaches, has already been applied to develop genetically modified therapeutic bacteria as pharmaceutical agents[8], such as the biocontainment of *Lactococcus lactis* by the removal of essential genes, such as *thyA*[19]. The *thyA* gene encodes thymidylate synthase, an enzyme required for the synthesis of deoxythymidine monophosphate, which is converted to deoxythymidine triphosphate essential for DNA synthesis. Deficiency of *thyA* causes thymineless death unless thymine or thymidine is added to the growth media[19]. However, the thymidine-auxotrophic system has the potential risk that, depending on the genetic characteristics of the surrounding microbial community, the engineered organisms might acquire the essential gene from other bacteria by HGT[20] and escape their containment. Additionally, the dissemination of synthetic gene circuits to wild-type (WT) strains in nature is still a concern, even though gene flow barriers have been developed by introducing toxin-antitoxin gene systems onto artificial plasmids and bacterial chromosomes[21]. Genes on a plasmid are frequently transferred among bacteria; such transfer is a major cause of the prevalence of antibiotic-resistant bacteria[22]. Existing containment systems can be greatly improved to prevent genetically engineered microorganisms and synthetic genes from spreading.

*Bacteroides thetaiotaomicron* is an attractive cellular chassis for therapies in the human gut microbiota[23]. This species stably colonizes the mammalian gut and is prevalent and abundant in the human gastrointestinal tract[24,25]. It has a broad ability to interact with its host as well as various other microbes because it ferments polysaccharides and glycans[26] and metabolizes bile salts[27]. *Bacteroides* spp. have been reported as an effective treatment for colitis[28,29], and genetic tools have already been developed to manipulate these cells to sense, record, and respond to a stimulus in the mammalian intestine[30–32]. However, the many potential applications of *B. thetaiotaomicron* have been stymied by the absence of a robust biocontainment system.

We have developed a biocontainment system in *B. thetaiotaomicron* that combines thymidine auxotrophy, an Engineered Riboregulator (ER) for controlled gene expression by the intended bacteria, and a CRISPR Device (CD) to achieve the reliable containment of engineered strains. This system can degrade an essential gene that could be transferred into the engineered bacteria from other bacteria and can also prevent engineered genetic circuits from disseminating to environmental bacteria. In addition to demonstrating the functionality of this biocontainment system, we provide validation of its evolutionary stability in vitro and in vivo.

We used a two-step methodology to construct a genetically modified, human commensal thymidine-auxotrophic bacterium bearing the functions mentioned above by introducing a gene cassette into the bacterial genome via recombination. This method, which can also be applied to a range of human gut-associated bacteria, enables the biological containment system to function stably with the controlled expression of a gene of interest (GOI), thus expanding future clinical applications of genetically engineered bacteria.

## Results

### Biological containment design

We developed a Cas9-assisted biological containment system that combines thymidine auxotrophy, an ER, and a CD to achieve controlled gene expression and biocontainment in engineered bacteria, and to prevent the release of the genetic circuit to other bacteria in the same environment (Fig. 1). The ER controls the level of gene expression of a GOI that is naturally repressed in all bacteria other than the engineered strain. Only the genetically modified strain can express the GOI; this strain has an intact ER, composed of trans-activating RNA (taRNA) and cis-repressed mRNA (crRNA) containing cis-repressive sequences (CRs) complementary to the taRNA. Without any taRNA, the crRNA solely represses the expression of the GOI[33], which could be introduced by genetic transduction via bacteriophages. The level of gene expression is controllable by optimization of the ER structure. The CD has been reported as a sequence-specific bactericidal device that induces double-stranded breaks in a target DNA sequence[34]. This system can prevent both the acquisition of an essential gene (*thyA* gene) from environmental bacteria by degrading the gene, and the dissemination of transgenes to environmental bacteria by killing the bacteria harboring the artificial gene cassette with the CD. The containment system provides controlled expression, reducing the potential risk of loss of auxotrophy and dissemination of transgenes.

The containment system was generated through recombination of plasmids bearing the ER and the CD (Supplementary Fig. 1) into the genome of *B. thetaiotaomicron*. Upstream and downstream sequences of the *thyA* gene were introduced flanking the gene cassette, such that double crossover recombination would lead to the integration of the ER and CD and the deletion of the *thyA* gene. The taRNA, SpCas9 gene, CR, and NanoLuc gene as a reporter of the GOI were introduced to the WT *B. thetaiotaomicron* VPI-5482 strain as the first step, and single guide RNA (sgRNA) targeting the *thyA* gene was integrated as the second step (Supplementary Fig. 2a). We then evaluated the characteristics of NanoLuc-producing genetically modified *B. thetaitaomicron* with the containment system.

### Repression and control of gene expression by the Engineered Riboregulator (ER)

The ER is composed of two components: the crRNA and the taRNA. The secondary structure in the 5' untranslated region (UTR) of the crRNA, which has a cis-repressive sequence (CR), represses the expression of the downstream gene by blocking access to the ribosome binding site (RBS). Sequences of taRNA containing a trans-activating sequence (TA) were optimized to hybridize to the crRNA and expose the RBS, initiating the translation of the NanoLuc reporter gene (Fig. 2a).

We first sought to build the crRNA structure to highly repress the expression of NanoLuc by inserting four types of CRs: crRCN, crR7N, crR10N or crR12N, into the RBS sequence (Supplementary Table 1). Previously, crRNAs had been constructed in *Escherichia coli* that

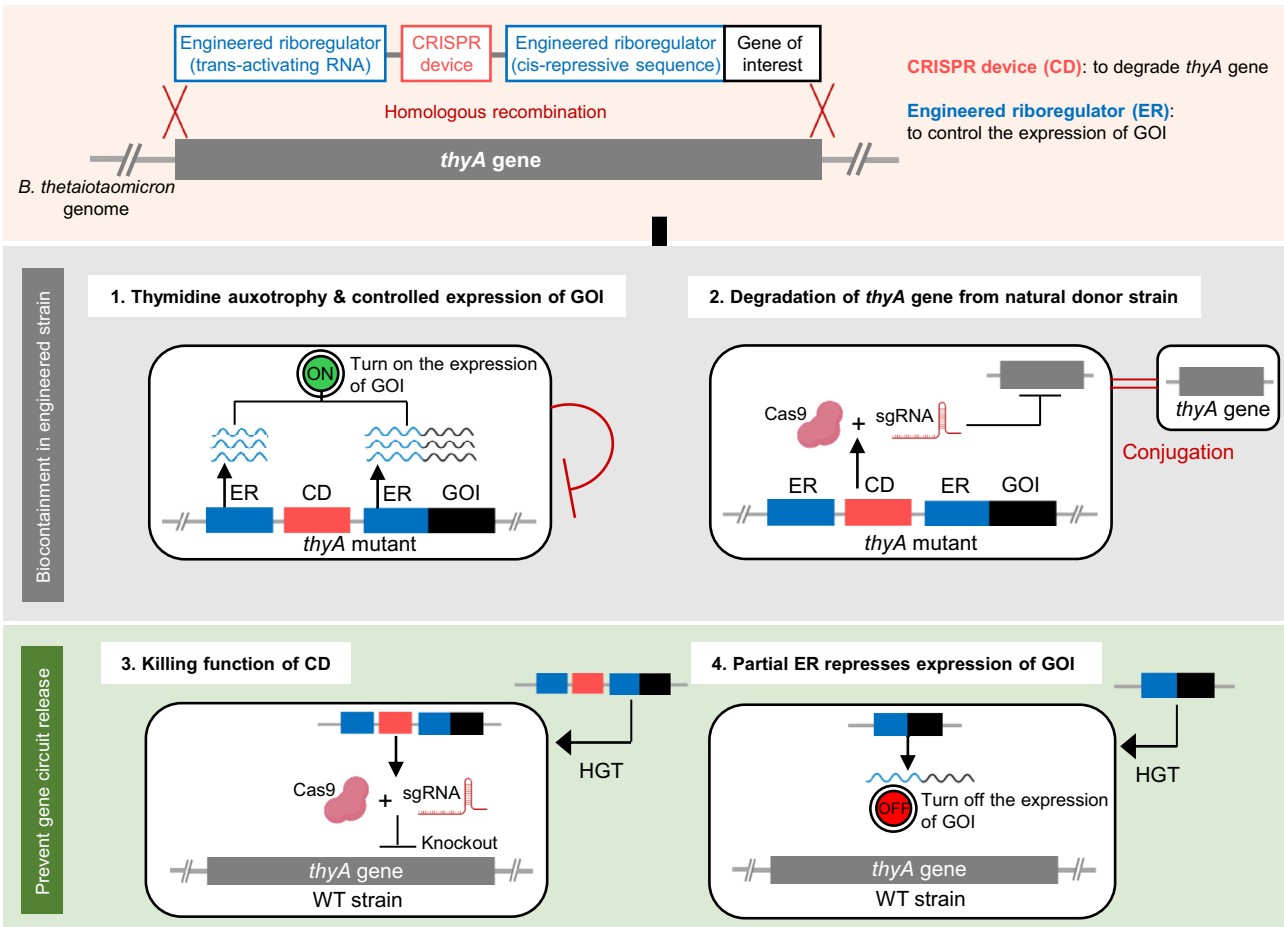

**Fig. 1 | Approach for biocontainment of the genetically modified strain and genetic elements.** Engineered Riboregulator (ER), CRISPR Device (CD), and Gene of Interest (GOI) are integrated into the genome of *B. thetaiotaomicron* by recombination of the sequence encoding these elements with the *thyA* gene. This sequence plays a role in biocontainment as follows: 1. Thymidine auxotrophy prevents the engineered strain from growing in the absence of thymidine. The ER controls the expression of the GOI. 2. The CD degrades the *thyA* gene from the donor strain, maintaining thymidine auxotrophy. 3. The released CD kills the wild-type (WT) strain, preventing the dissemination of the gene circuit by knocking out the essential *thyA* gene in the genome. 4. When released to the WT strain, the partial ER represses the expression of the GOI. HGT horizontal gene transfer.

blocked the purine-rich RBS region upstream of the start codon by the formation of secondary structures[33]. Unlike *E. coli*, in which the homology of the Shine-Dalgarno sequence to the 16 S rRNA largely dictates the strength of translation, the strength of gene expression in *Bacteroides* correlates poorly with the level of RBS complementarity to the 16 S rRNA[35,36]. The RBS of *Bacteroides* tends to be AT-rich and more sensitive to secondary structure than that of *E. coli*[31]. The putative S1 protein binding site in the RBS was reported to have a strong effect on the strength of the RBS[37]. Therefore, we originally targeted this region to make crRNA structures. To design the ER, we used RNAfold WebServer, a server designed for the analysis of nucleic acid systems[38]. The generated RNAfold-predicted structures of crRNA showed various states of the putative S1 protein binding site with base-pair probabilities ranging from zero to one (Fig. 2b).

To evaluate the degree of repression of the GOI by crRNA, we tested the intensity of luminescence through enzymatic activity of NanoLuc. All strains except for the WT strain had luciferase activity (Fig. 2c). However, the activities of those which carried crRCN or crR7N were significantly repressed: 496-fold or 253-fold, respectively, compared to the strain without a CR (rpiL*). This repression would be caused by the secondary structures of crRNA which block the putative S1 protein binding site, and their higher base-pair probabilities. The rpiL* strain showed much higher output than crR10N even though the putative S1 protein binding site was blocked. Compared to crR10N

(green color), the secondary structure of rpiL* (blue color) had the lowest probability of base-pairing (Fig. 2b), and it is possible that the rpiL* might form a different open structure inside a cell.

Seven types of taRNA were then designed, by introducing inner loops and bulges to destabilize the structures, so that they had different Minimum Free Energy (MFE) and length complementary to crRNA (Supplementary Table 2). We thought that, in addition to the MFE, the mismatches would contribute to resistance to RNAse III cleavage of RNA duplexes[33]. The structures of taRNA were analyzed using RNAfold WebServer (Fig. 2d), and the outputs of NanoLuc were measured. NUPACK web server was also utilized for the crRNA/ taRNA complex. As shown in Fig. 2e and Supplementary Fig. 3a, crRNA/taRNA complexes had exposed putative S1 protein binding sites. crRCN and crR7N significantly repressed the expression of NanoLuc in the absence of any taRNA, while with taRNA, the NanoLuc activity of the strain that had taRNA6 (crRNA-taRNA6) significantly increased (15 fold; Fig. 2f). The stability of the taRNA affected gene expression, given that the MFE of taRNAs highly correlated with the outputs whereas the stability of the crRNA/ taRNA complex did not show any correlation with gene expression levels (Fig. 2g; Supplementary Fig. 3b).

The outputs of this ER system are likely to increase at temperatures higher than the physiological one because the MFE of both taRNAs and crRNAs would rise, and vice versa (outputs would decrease at lower-than-physiological temperatures). In addition to the

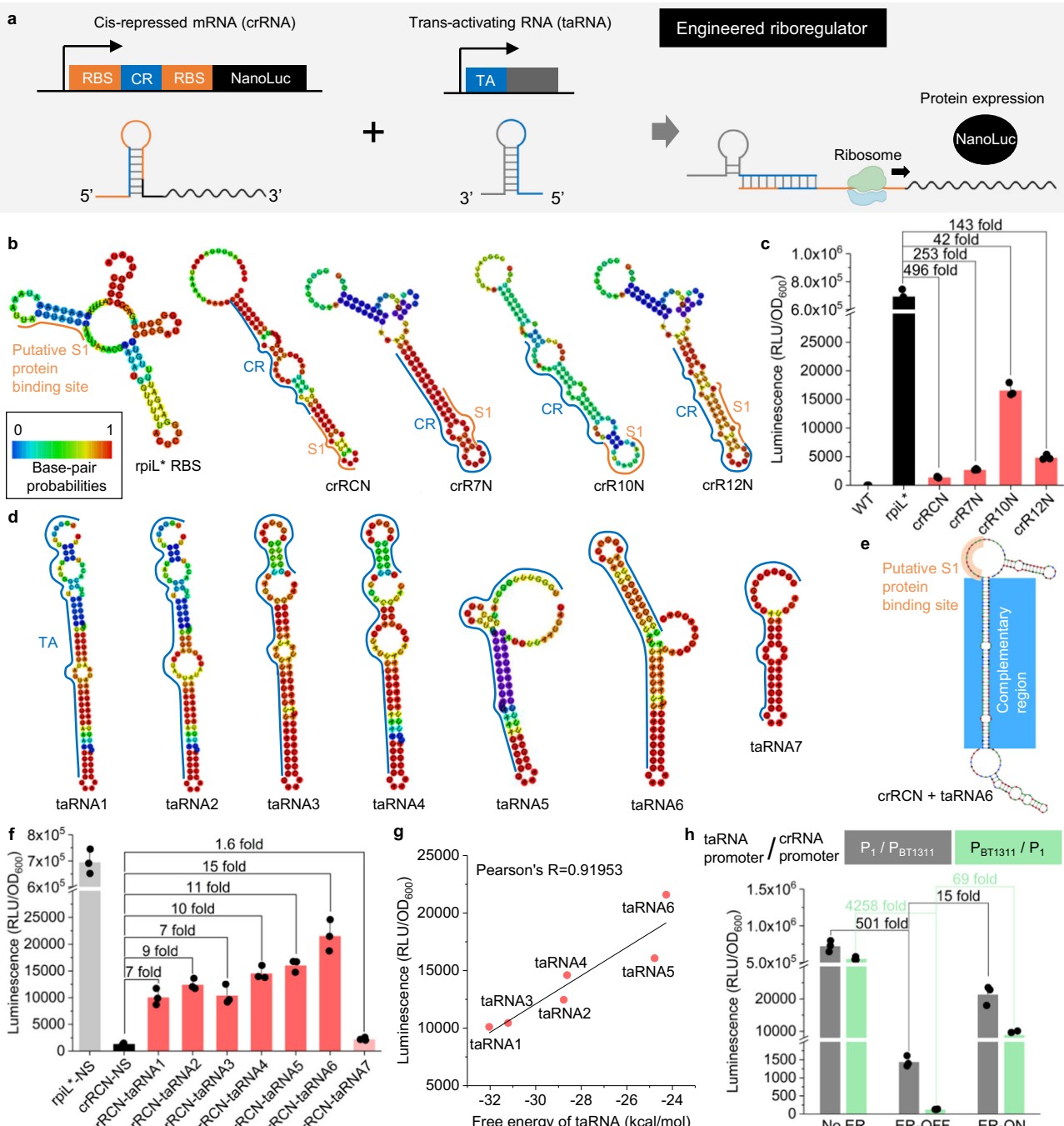

**Fig. 2 | Engineered riboregulator (ER) system used to control the expression of the gene of interest in genetically modified *Bacteroides thetaiotaomicron*.**
**a** Schematic of ER system. CR and TA represent cis-repressive and trans-activating sequences, respectively. **b** RNAfold-WebServer-predicted structures of cis-repressed mRNA (crRNA) variants. **c** Results of cis repression of crRNA variants. The luminescence values were normalized to $OD_{600}$ of the cell suspensions. Fold changes were calculated on the basis of the value with no CR (rpiL*) and non-sense taRNA. **d** RNAfold-WebServer-predicted structures of trans-activating RNA (taRNA) variants. **e** NUPACK-predicted structure of crRNA (crRCN)/taRNA6 complex. **f** Outputs of NanoLuc for taRNA variants. NS represents non-sense taRNA.

**g** Correlation between outputs and free energy of folding for taRNA variants: taRNA1 to taRNA6. **h** ON/OFF outputs with $P_{BT1311}$ and $P_1$ promoters switched between crRNA and taRNA. Gray columns represent original promoters ($P_1$/$P_{BT1311}$) whereas green ones show switched promoters ($P_{BT1311}$/$P_1$). ER-OFF, with crRCN and NS; ER-ON, with crRCN and taRNA6. (**c**, **f**, **h**) Dots are three biological replicates made on different days and columns are the values of the mean. Error bars represent the standard deviations of the three biological replicates. **g** Dots show the luminescent values of the mean and Minimum Free Energy of taRNA. Source data are provided as a Source Data file.

temperature response of this ER system, a longer complementary sequence than that of 48 bps was required to activate the crRNA. Weaker activation of crRCN by taRNA7 was observed, a taRNA which was designed so that it would have a shorter region of complementarity (26 bps) than the others (48 bps). In contrast, the

resistance of RNA duplexes to RNase III cleavage was not critical to the output, though taRNA3 was built to test resistance by adding mismatches to the RNA/ RNA complex.

To further improve the ON/ OFF ratio of the gene expression level, we switched the promoters ($P_{BT1311}$ or $P_1$) controlling taRNA and crRNA

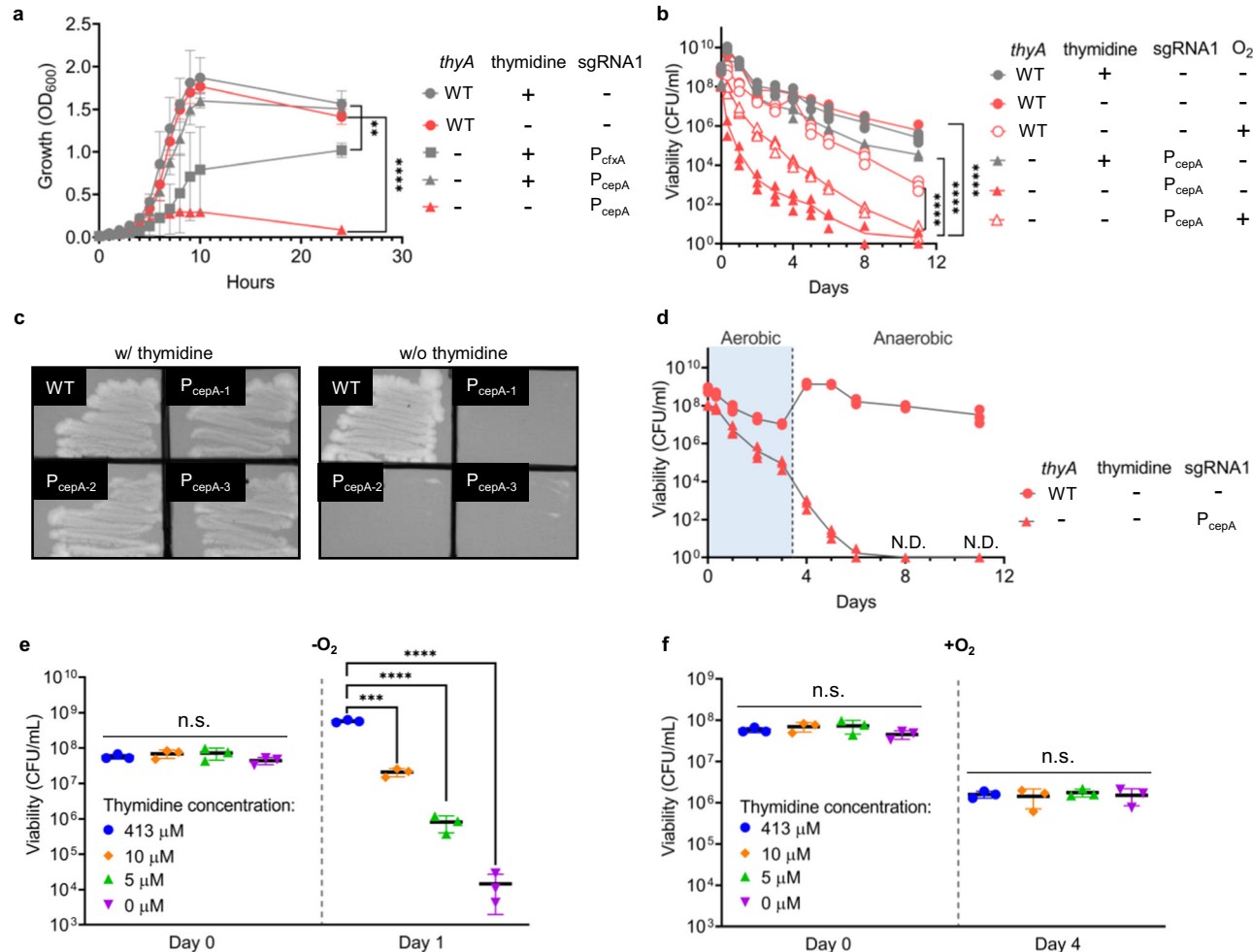

**Fig. 3 | Thymidine auxotrophy of genetically modified *B. thetaiotaomicron*.**
**a** Growth curves of wild-type (WT) and *thyA*-deficient strains with $P_{cfxA}$-sgRNA1 and $P_{cepA}$-sgRNA1 in the presence (413 μM) and absence of thymidine. **b** Cell viability of WT and *thyA*-deficient strain with $P_{cepA}$-sgRNA1 in the presence (413 μM) and absence of thymidine under anaerobic and aerobic conditions. **c** Restreaked colonies of the *thyA*-deficient strain with $P_{cepA}$-sgRNA1 on TYG with thymidine (413 μM) and without thymidine, after anaerobic culturing without thymidine for 11 days. **d** Cell viability of WT and *thyA*-deficient strain with $P_{cepA}$-sgRNA1 when cells were transferred from aerobic to anaerobic conditions in the absence of thymidine. **e** Cell viability of *thyA*-deficient strain with $P_{cepA}$-sgRNA1 at various concentrations of thymidine in the anaerobic condition. **f** Cell viability of the *thyA*-deficient strain

with $P_{cepA}$-sgRNA1 at various concentration of thymidine in the aerobic condition. **a** Each dot is the value of the mean, and error bars represent the standard deviations of three biological replicates made on different days (Tukey's multiple comparisons test with one-way analysis of variance on data at 24 h; *thyA⁻* with thymidine and $P_{cfxA}$-sgRNA1 vs. *thyA⁻* with thymidine and $P_{cepA}$-sgRNA1: $P = 0.0010$, ***P < 0.01** ****$P < 0.0001$). **b–f** Each dot is a biological replicate ($n = 3$), and lines and central bars are the values of the mean. Error bars represent standard deviations (Tukey's multiple comparisons test with one-way analysis of variance on log-transformed data; **e** 413 μM vs. 10 μM at Day1: $P = 0.0005$, ***P < 0.001** ****$P < 0.0001$). The detection limit is 1 CFU/ml. n.s. represents not significant. N.D. represents no detection. Source data are provided as a Source Data file.

expression (Supplementary Fig. 2a). $P_{BT1311}$ has been reported to be a stronger promotor than $P_1$[31]. We expected gene expression to be repressed more strongly without complementary taRNA (OFF state) if the weaker promoter was placed upstream of crRNA, whereas we expected that the expression level would be retained with taRNA6 (ON state) if the stronger promoter was placed upstream of the taRNA sequence. As shown in Fig. 2h, the strain bearing the switched promoters ($P_{BT1311}$/ $P_1$) had a remarkably improved ON/OFF ratio. This taRNA elicited a ~70-fold change in gene expression between the ON and OFF states.

#### Effect of CRISPR Device (CD) on cell growth

We next investigated whether the CD, composed of the SpCas9 gene and sgRNA, affected the cell growth of the genetically modified thymidine-auxotrophic *B. thetaiotaomicron* strain. Two sets of sgRNAs and promoter sequences ($P_{cfxA}$-sgRNA1 and $P_{cepA}$-sgRNA1) were introduced, via the second double crossover, into the

strain having crRCN and taRNA6 as an ER (Supplementary Fig. 2). $P_{cfxA}$-sgRNA1 targets the *thyA* gene with the stronger promoter $P_{cfxA}$. In fact, the $P_{cfxA}$ promoter is approximately 10-fold stronger than the $P_{cepA}$ promoter[31].

We next tested the growth curves of the strains in the presence and absence of thymidine. With thymidine, the $P_{cfxA}$-sgRNA1 strain grew more slowly than the other strains (Fig. 3a), possibly because the high concentration of sgRNA was harmful to this strain. The high concentration of the sgRNA might have increased the frequency of small-RNA-mediated targeting of mRNA[39,40]. On the other hand, we also observed that with thymidine the $P_{cepA}$-sgRNA1 strain grew rapidly, at the rate close to that of the WT strain, but without thymidine this strain grew just a little and then died; the initial growth would have been enabled by the thymidine retained in the cells before the assay began. Overall, high expression of this sgRNA can lead to toxicity and inhibition of cell growth.

## Viability of genetically modified thymidine-auxotrophic *B. thetaiotaomicron*

To validate thymineless death as a method of biological containment for the genetically modified strain, we measured cell viability anaerobically and aerobically. Under anaerobic conditions, the cell concentration of the WT strain increased to over $1 \times 10^9$ CFU/mL within 8 h and then gradually decreased as a result of lack of nutrients or the accumulation of waste products. Nevertheless, the cell concentration remained higher than $1 \times 10^5$ CFU/mL for 11 days. Likewise, the $P_{cepA}$-sgRNA1 strain with thymidine showed large numbers of living cells throughout the culture (Fig. 3b). On the other hand, we observed an approximately $10^4$-fold decrease in CFU/mL of the genetically modified strain without any sgRNA and a $10^6$-fold decrease in the $P_{cepA}$-sgRNA1 strain 6 days after incubation without thymidine (Supplementary Fig. 4). The CD seemed to help increase the death rate, which might be derived from its toxicity. When the culturing was prolonged up to 11 days, the CFU/mL of $P_{cepA}$-sgRNA1 strain decreased by $10^7-10^8$ (Fig. 3b). We streaked three colonies of the strain obtained at day 11 onto a TYG (trypticase-yeast extract-glucose) agar plate with and without thymidine to determine whether the viable cells were still thymidine auxotrophs. As shown in Fig. 3c, these cells showed thymidine auxotrophy and would die without any growth even if the viability test was extended after day 11.

When the $P_{cepA}$-sgRNA1 strain was cultured aerobically, the rate of decrease in viable cells was repressed (Fig. 3b). The cell growth, or DNA replication, of this anaerobe is inhibited in the aerobic state, which would have repressed the thymineless death[41]. This result was consistent with a previous report about *Bacteroides ovatus*[20]. However, our strain decreased in cell numbers more rapidly than the WT strain and did not grow any more, even if it was transferred from the aerobic state to the anaerobic state (Fig. 3b, d).

Viability was tested at various concentrations of thymidine to clarify the concentration dependency of thymineless death (Fig. 3e, f). The decrease in viable cells was not caused by thymineless death in the aerobic state, but possibly by stress from the sgRNA and/or oxygen. On the other hand, under anaerobic conditions, this strain died rapidly at concentrations of thymidine less than 5 μM.

The inhibition of thymineless death was also observed when the $P_{cepA}$-sgRNA1 strain was subjected to other stressful conditions. To test the effect of varying the carbon source, we measured the growth curve and viability of the engineered *B. thetaiotaomicron* in the presence of monosaccharide (i.e., glucose and galactose), disaccharide (i.e., maltose), and polysaccharide (i.e., starch), with or without thymidine (Supplementary Fig. 5a, b). Engineered *B. thetaiotaomicron* grew rapidly in the presence of thymidine with either a monosaccharide or a disaccharide as the carbon source. On the contrary, no viable cells were detected in the absence of thymidine after 48 h of incubation with these carbon sources (Supplementary Fig. 5b). The growth of the strain was much slower when starch was used as the sole carbon source with thymidine than when a monosaccharide or a disaccharide was used (Supplementary Fig. 5b). With starch as the sole carbon source, after 96 h of incubation in the absence of thymidine significantly lower numbers of viable cells were detected, compared to those with thymidine, (Supplementary Fig. 5b).

We also tested the effect of carbon source starvation on the auxotrophy in the engineered *B. thetaiotaomicron* (Supplementary Fig. 5c). Bacterial cells were cultured in a defined minimal medium (DMM) without glucose after a PBS wash in the presence or absence of thymidine. As expected, the cells stopped growing without a carbon source. Moreover, the viability of strains in the absence of thymidine significantly decreased after 48 and 96 h of incubation, indicating that thymidine auxotrophy persisted in conditions of carbon source starvation.

Considering the important roles and ubiquitous presence of bile acids in the human gut, we tested the effect of crude bile acid and secondary bile acid (i.e., deoxycholic acid, or DCA) on the auxotrophy of engineered *B. thetaiotaomicron* (Supplementary Fig. 5d, e). The bacteria stopped growing without thymidine (Supplementary Fig. 5d), and no viable cells could be detected after 48 h of incubation in the absence of thymidine in TYG medium supplemented with either crude bile acid or DCA (Supplementary Fig. 5e).

In summary, as the levels of carbon sources and bile acids fluctuate in the gut, their presence or absence can contribute to conditions of stress. Although one carbon source (starch) could delay thymineless death, our engineered strain of *B. thetaiotaomicron* nevertheless maintained its auxotrophy, indicating the robustness and efficiency of our biocontainment.

## Blocking acquisition of *thyA* gene by genetically engineered bacteria

Next, we tested whether the CD would prevent the genetically engineered strain from acquiring the *thyA* gene from another bacterial strain. When the *E. coli* S17-1 λ pir strain, carrying a plasmid bearing the intact or mutated *thyA* gene, was mixed with the genetically modified *B. thetaiotaomicron* strain, the plasmid transferred to the *Bacteroides* strain by conjugation (Fig. 4a).

To determine whether the CD carried by the genetically modified strains of *B. thetaiotaomicron* would degrade the acquired *thyA* gene, several genetically modified *Bacteroides* strains with different ERs and CDs were constructed and tested for their ability to prevent the acquisition of the *thyA* gene. One strain had neither ER nor sgRNA; others had the ER with or without sgRNAs, including the non-targeting control. Plasmids bearing either an intact or a mutated *thyA* gene were generated by introducing synonymous codons into the sgRNA1-targeting sequence. Filter mating between *B. thetaiotaomicron* and *E. coli* with the plasmids was performed on TYG agar plates in the absence of thymidine.

As shown in Fig. 4b, *B. thetaiotaomicron* transconjugants were detected on BHIS agar plates with erythromycin and thymidine, indicating that these strains had acquired the plasmid-borne mutated *thyA* gene that was not targeted by a CD. Additionally, the strain with the CD having the non-targeting control, which did not degrade the *thyA* gene, had obtained the intact *thyA* gene and grew on the BHIS plates. In contrast, no colonies were observed of the bacterial strain having the CD that targeted the intact, plasmid-borne *thyA* gene. This result demonstrated that the CD can specifically recognize and destroy *thyA*, blocking the effect of HGT of the plasmids. The ability to disable the gene that would otherwise correct the auxotrophy after HGT has occurred is a requirement for safe biological containment.

We also conducted a conjugation experiment on TYG agar plates in the presence of thymidine. Some viable erythromycin-resistant cells having sgRNA1 were detected, in contrast to results obtained in the filter mating without thymidine. These cells would be derived from the higher numbers of viable cells during conjugation compared to the conjugation in the absence of thymidine, which leads to higher numbers of transconjugants. Nevertheless, the conjugation efficiency of the plasmid having an intact *thyA* gene was significantly decreased, at least 156-fold, by the CD (Fig. 4c). The colonies selected with an erythromycin resistance gene were still thymidine auxotrophic and did not have any *thyA* gene since no amplicon was detected by PCR using primers binding to the *thyA* gene (Supplementary Fig. 6a, b). Whole genome sequencing revealed that the backbone of the plasmid bearing the intact *thyA* gene (pNH9223) had integrated downstream of the NanoLuc gene without any integration of *thyA* gene in the transconjugant (Supplementary Fig. 6c; BioSample accession number: SAMN20797097). Therefore, the actual frequency of *thyA* gene transfer would be less than the frequency of erythromycin resistance gene transfer. These results also indicated that the CD could also repress the acquisition of the genes adjacent to the targeted genes.

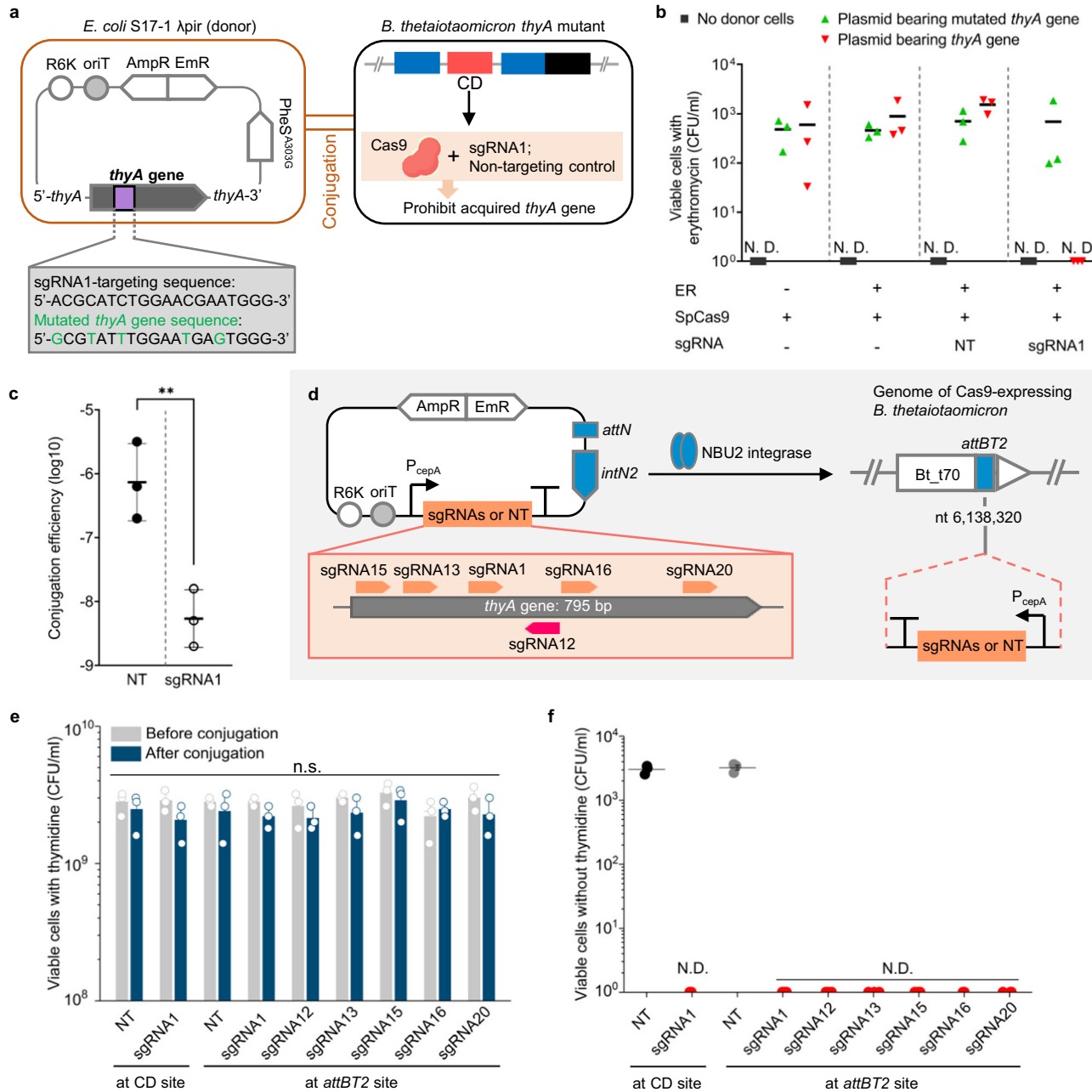

**Fig. 4 | Prevention of acquisition of *thyA* gene by genetically engineered strain.** **a** Schematic of plasmid construct bearing the *thyA* gene and Horizontal Gene Transfer (HGT) of the *thyA* gene by conjugation from *E. coli* to genetically engineered thymidine-auxotrophic *B. thetaiotaomicron*. The *E. coli* strains have either an intact or a mutated *thyA* gene on their plasmids. The *B. thetaiotaomicron* strains have either the sgRNA1 sequence or a non-targeting control (NT). R6K origin of replication, oriT origin of transfer, AmpR ampicillin resistance cassette, EmR erythromycin resistance cassette, PheS[A303G], mutated α−subunit of phenylalanyl-tRNA synthetase gene; *intN2*, tyrosine integrase cassette; *attN*, recombination site with *attBT2* sites on the genome of *B. thetaiotaomicron* **b** Viable *B. thetaiotaomicron* cells following conjugation on TYG agar plates without thymidine and plating on BHIS agar plates with gentamicin, thymidine, and erythromycin. **c** Conjugation efficiency of *thyA* gene through conjugation on TYG agar plates containing thymidine. The efficiency was calculated by dividing CFU/ml on the BHIS agar plates with

gentamicin, thymidine, and erythromycin by total CFU/ml (two-tailed unpaired Student's t-test on log-transformed data; NT vs. sgRNA1: $P = 0.0080$, **$P < 0.01$). **d** Schematic of integration of a plasmid bearing each of seven types of sgRNA sequences, including NT on the genome of SpCas9-expressing *B. thetaiotaomicron* by pNBU2 technology. **e** Cell viability of *Bacteroides* before and after conjugation between the *E. coli* and genetically engineered thymidine-auxotrophic *B. thetaiotaomicron* with various types of sgRNAs (two-tailed unpaired Student's t-test on log-transformed data between before and after conjugation). **f** Viable *B. thetaiotaomicron* when cells were plated on TYG agar plates without thymidine after conjugation on TYG agar plates with thymidine. Three biological replicates were made on the different days. Black bars and gray columns are the values of the mean. Error bars are the standard deviations. The detection limit is 1 CFU/ml. n.s. represents not significant. N.D. represents no detection. Source data are provided as a Source Data file.

Next, we sought to clarify whether the capability to prevent the HGT of the *thyA* gene depends on the targeted sequences in the gene. Genetically modified thymidine-auxotrophic strains bearing the CD were constructed by integration of pNBU2-based plasmids with sgRNAs

and the erythromycin resistance gene after the first recombination[31]; these strains each carried one of the seven types of the sgRNAs (Fig. 4d; Supplementary Table 3) and their promoter $P_{cepA}$ at one of two *attBT2* sites located in the 3′-ends of the two tRNA[Ser] genes, BT_t70 and BT_t71,

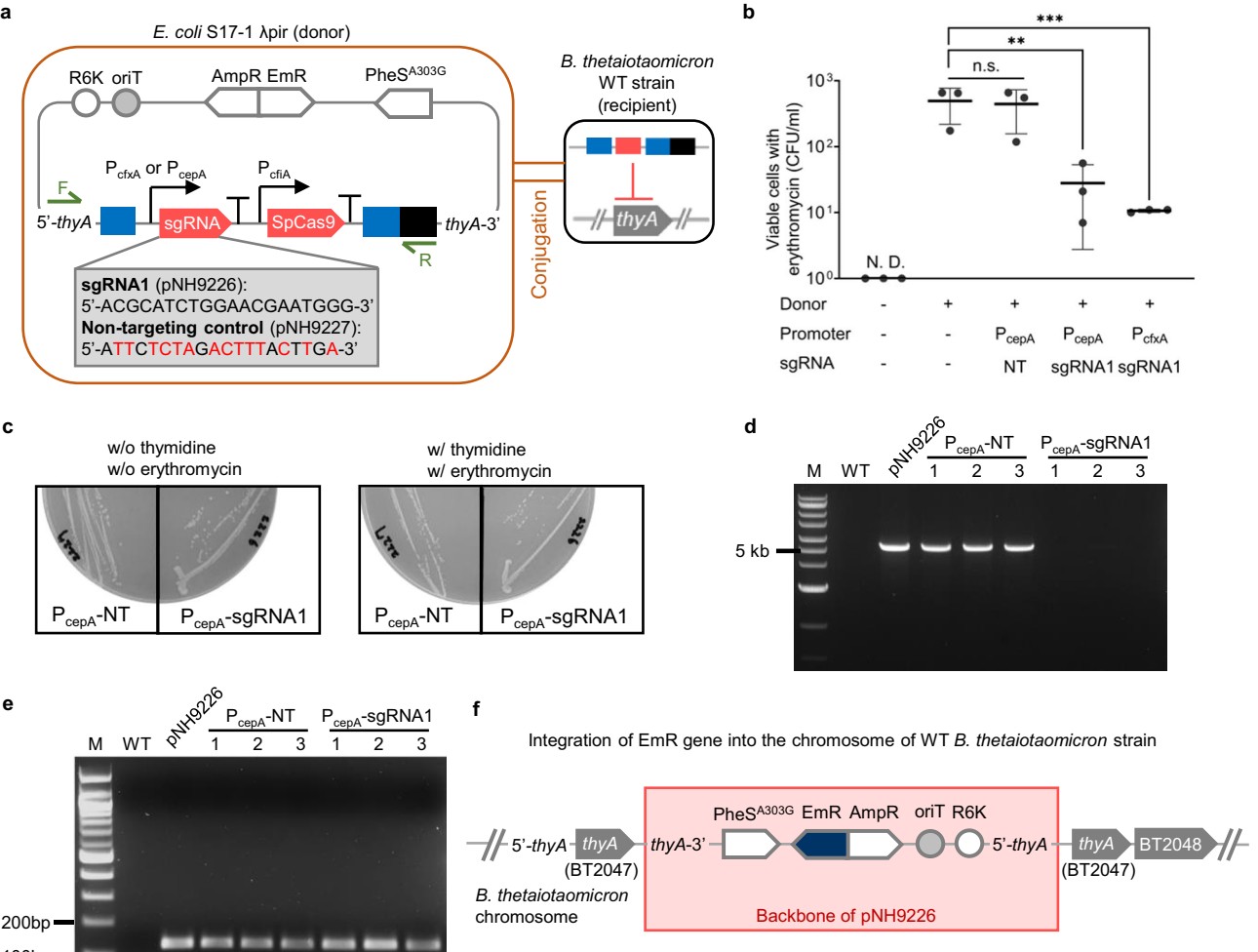

**Fig. 5 | Prevention of dissemination of transgenes by genetically engineered strain. a** Schematic of plasmid construct bearing transgenes and Horizontal Gene Transfer (HGT) of this plasmid by conjugation from *E. coli* to wild-type (WT) *B. thetaiotaomicron*. The *E. coli* strains have either sgRNA1 or a non-targeting control (NT) on their plasmids. The *B. thetaiotaomicron* strain has an intact *thyA* gene. R6K, origin of replication; oriT, origin of transfer; AmpR, ampicillin resistance cassette; EmR, erythromycin resistance cassette; PheS[A303G], mutated α−subunit of phenylalanyl-tRNA synthetase gene **b**, Viable *B. thetaiotaomicron* when cells were plated on BHIS agar plates with gentamicin, thymidine, and erythromycin after conjugation on TYG agar plates without thymidine. **c** Restreaked colonies on TYG with and without thymidine and erythromycin. **d** Amplicon patterns of three transconjugants isolated from TYG agar plates with thymidine and erythromycin

after conjugation. PCR was performed with primers shown in (**a**), which are named Seq-thyA-F and mmD663 in Supplementary Table 6. **e** PCR amplification of EmR gene fragment (144 bp) in transconjugants with primers named qPCR-EmR-F and qPCR-EmR-R in Supplementary Table 6. **f** Integration of pNH9226 backbone into the chromosome of WT *B. thetaiotaomicron* strain. Three biological replicates were made on the different days. Black bars are the values of the mean. Error bars are the standard deviations (Tukey's multiple comparisons test with one-way analysis of variance on log-transformed data; *E. coli* having P$_{cepA}$-sgRNA1, and *E. coli* having P$_{cfxA}$-sgRNA1 vs. *E. coli* without any promoter or sgRNA: $P = 0.0028$, and $P = 0.0007$, **$p < 0.01$, ***$p < 0.001$). The detection limit is 1 CFU/ml. n.s. represents not significant. N.D. represents no detection. Source data are provided as a Source Data file.

on the *B. thetaiotaomicron* chromosome. These strains showed thymidine auxotrophy and antibiotic resistance when they were plated on gentamicin-containing TYG agar plates either with thymidine and erythromycin, or without either (Supplementary Fig. 7). *B. thetaiotaomicron* strains with different *thyA*-targeting sgRNAs displayed similar viability in the presence of thymidine before and after conjugation (Fig. 4e). The filter-mating experiment using thymidine auxotrophy as a selective marker revealed that all the strains had the capacity to avoid the HGT of the *thyA* gene (Fig. 4f). The choice of targeted sequence did not seem to highly affect the capability to degrade the *thyA* gene. Furthermore, the CD worked well even though the sgRNAs were integrated at a locus far from the SpCas9 gene. Thus, there seems to be no requirement to target specific sequences of the *thyA* gene in this containment system or to integrate sgRNAs at specific sites, though further research on the effect of the targeted sequence and sgRNA site on disabling the *thyA* gene is still needed.

## Prevention of dissemination of transgenes by genetically engineered bacteria

We evaluated whether the CD could block the HGT of transgenes, such as the CD, ER, and GOI, through conjugation between *E. coli* and *B. thetaiotaomicron*. The donor strain, *E. coli* S17-1 λ pir, which carries a plasmid bearing the transgenes (shown in Fig. 5a), was mixed with WT *B. thetaiotaomicron*, the recipient. The plasmids with the CD were expected to degrade the *thyA* gene on the recipient genome so that transconjugants would not survive.

When we selected the transconjugants on BHIS agar plates with gentamicin, thymidine, and erythromycin, the number of transconjugants that had received the plasmid with the CD containing P$_{cepA}$ and P$_{cfxA}$ was markedly lowered, by approximately 18 and 46 times, respectively, compared to that without either promoter or sgRNA (Fig. 5b). This result indicated that the *thyA* gene on the recipient genome had been destroyed by the CD and that most of the recipients that

had gotten the plasmid died because their genome was not repaired. The stronger promoters are utilized for the transcription of sgRNA1, the more strictly the HGT is likely to be regulated, as fewer viable cells with $P_{cfxA}$ were detected on the BHIS agar with erythromycin than cells with $P_{cepA}$, though the strength of the promoters did affect cell growth, as described. The appropriate choice of promoter can, therefore, be helpful to avoid the dissemination of the transgenes.

When the colonies of transconjugants were selected with erythromycin and restreaked on TYG in the presence or absence of both thymidine and erythromycin, they showed erythromycin resistance and thymidylate synthase activity regardless of the sgRNAs (Fig. 5c). Thus, the transconjugants with $P_{cep}$-sgRNA1 had acquired the erythromycin resistance gene while retaining the *thyA* gene. To clarify why these transconjugants had an intact *thyA* gene after transfer of the plasmid bearing the erythromycin resistance gene and the CD, the region of transgenes in the transconjugant genomes was amplified by PCR using the primers in Fig. 5a. Interestingly, the DNA purified from transconjugants obtained by conjugation with the *E. coli* strain having $P_{cepA}$ and sgRNA1 did not have the transgenes, whereas transgenes containing $P_{cepA}$ and the non-targeting control were detected in the genome of WT *B. thetaiotaomicron* after conjugation (Fig. 5d). As PCR amplification showed, the recipients indeed had acquired the erythromycin resistance gene (Fig. 5e). Whole genome sequencing revealed that only the backbone of the plasmid with the erythromycin resistance gene, but not the CD, was integrated into the genome of the WT recipient strain (Fig. 5f; BioSample accession number: SAMN20797095). The recipient may have received the erythromycin resistance gene and released the transgenes containing the CD because this portion of the transferred plasmid was detrimental to the WT strain. Therefore, the actual frequency of transgenes transferred to the recipient strains would be lower than the value calculated as erythromycin-resistant transconjugants.

## Evolutionary stability of genetic elements

To evaluate stability, we tested how long the ER and CD remained stable during cell growth in vitro. The genetically modified strain with those functions was monocultured up to 21 days and checked for gene expression of NanoLuc by the ER and the prevention of HGT of the *thyA* gene by the CD every 7 days.

Over the time course of the evolutionary stability test, the expression of NanoLuc was maintained, resulting from the stability of the ER (Fig. 6a). The genetically modified *Bacteroides* strain could receive the plasmid bearing the mutated *thyA* gene, as colonies were detected throughout the period of this experiment. On the other hand, bacterial colonies that had obtained the plasmid having an intact *thyA* gene were not observed for the first 14 days. Viable erythromycin-resistant cells appeared at day 21 (Fig. 6b). Given that there was no erythromycin-resistant colony without any donor cells (the negative control) and there was no detection of the erythromycin resistance gene by PCR in the recipients before conjugation (Supplementary Fig. 8a), we can conclude that the viable cells were not derived from spontaneous mutation during the 21 days of culture. The integration of the plasmid bearing the erythromycin resistance and the intact *thyA* genes was observed by whole genome sequencing (Supplementary Fig. 8b; BioSample accession number: SAMN20797096). Therefore, the CD was stable for at least 14 days under these growth conditions. We assumed that some mutations may have occurred in the sequence of the CD during cell growth and that the CD may have lost functionality after day 14. However, Sanger sequencing revealed no mutation in the region of the CD for thirteen viable colonies grown on BHIS agar with gentamicin, thymidine, and erythromycin. The CD may have lost its functionality via another mechanism, such as a mutation at a locus outside the CD region, though this is still unclear. Overall, this engineered strain was stable for 14 days, which is equivalent to at least 240 cell divisions, corresponding to a $10^{72}$-fold amplification of the initial inoculant.

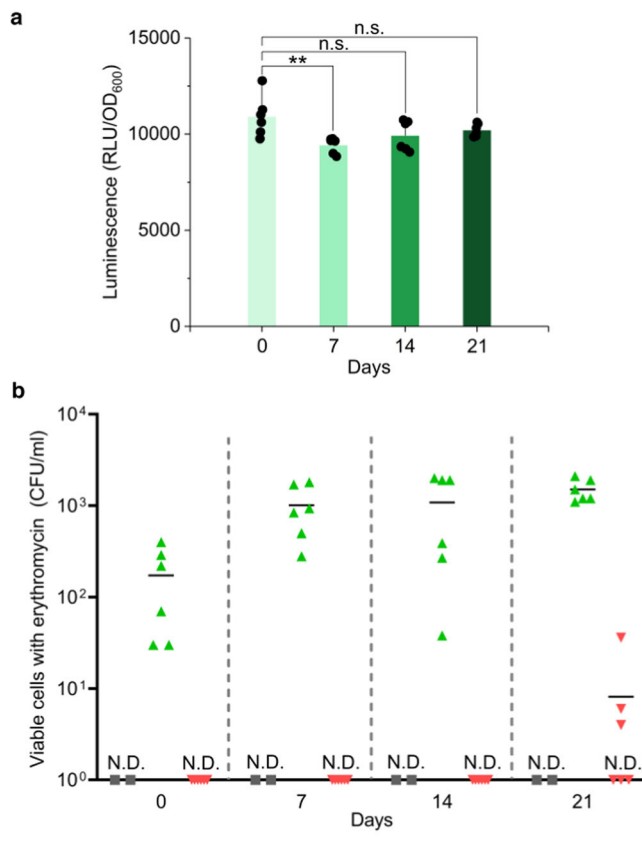

**Fig. 6 | In-vitro genetic stability of the genetically engineered strain bearing Engineered Riboregulator (ER) and CRISPR Device (CD). a** Outputs of NanoLuc measured every 7 days until day21. Three biological replicates were made twice. Dots are six biological replicates and columns are the values of the mean. Error bars represent the standard deviations of the six biological replicates (Tukey's multiple comparisons test with one-way analysis of variance on data; day0 vs. day7: $P = 0.0088$, $**P < 0.01$). **b** Viable *B. thetaiotaomicron* when cells were plated on BHIS agar plates with thymidine, erythromycin, and gentamicin every 7 days until day21 after conjugation on TYG agar plates without thymidine. Three biological replicates were made twice. Individual dots represent six biological replicates of recipients, whereas two biological replicates were made for No donor cells (control). Black bars are the values of the mean. The detection limit is 1 CFU/ml. n.s. represents not significant. N.D. represents no detection. Source data are provided as a Source Data file.

## In vivo colonization of the genetically modified *B. thetaiotaomicron* and function of the Engineered Riboregulator (ER), thymidine auxotrophy, and CRISPR Device (CD)

To evaluate whether the genetically modified thymidine-auxotrophic bacteria can colonize mice and whether the ER and thymidine auxotrophy function in vivo, three *B. thetaiotaomicron* strains were administered to specific-pathogen-free (SPF) mice: a *thyA*-deficient strain with the ER and CD, $\Delta thyA$ (ER⁺/CD⁺); a *thyA*-deficient strain with the ER and the non-targeting CD, $\Delta thyA$ (ER⁺/CD^{NT}); and a *thyA*-intact strain without either, *thyA*⁺ (ER⁻/CD⁻). Stools were collected for 10 days to quantify NanoLuc luminescence and viable cells (Fig. 7a). As shown in Fig. 7b, when mice were gavaged with either the $\Delta thyA$ (ER⁺/CD⁺) or the $\Delta thyA$ (ER⁺/CD^{NT}) strain after 7 days of the antibiotic treatment, the intensity of luminescence increased and then remained stable up to day 10. Thus, the genetically modified thymidine-auxotrophic strains had stably colonized the mice and the ER worked well in vivo. High concentrations of thymidine in the mouse small intestine, which have been reported to be on the order

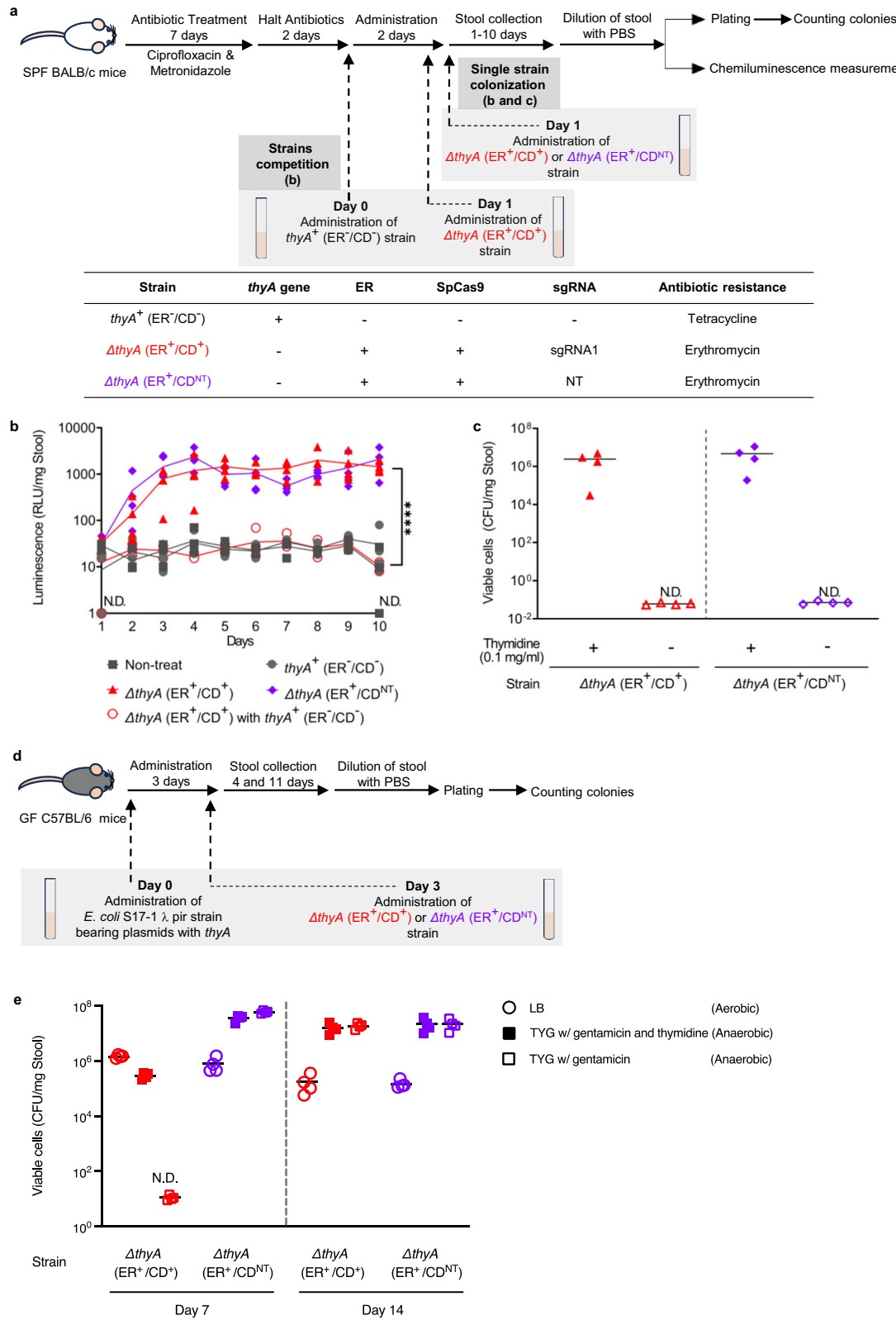

of several hundred μM[42], may have supported the colonization of the thymidine-auxotrophic strains.

A competition experiment between the *ΔthyA* (ER+/CD+) or the *thyA+* (ER−/CD−) strains was also performed. Once the *thyA*-intact strain *thyA+* (ER−/CD−) colonized after the antibiotic treatment, there was inhibition of colonization by *ΔthyA* (ER+/CD+), and the output of NanoLuc disappeared. The in vitro growth curve showed that *thyA+*

(ER−/CD−) grew faster than *ΔthyA* (ER+/CD+) (Supplementary Fig. 9). The difference in the growth rate, as well as the space occupied by *thyA+* (ER−/CD−) and the competition for nutrients, may have inhibited stable colonization by *ΔthyA* (ER+/CD+).

To test the function and stability of thymidine auxotrophy in vivo, viable cells in fecal pellets at day 10 from the *ΔthyA* (ER+/CD+)-mono-administered and *ΔthyA* (ER+/CDNT)-mono-administered mice were

**Fig. 7 | In vivo colonization, function of Engineered Riboregulator (ER), thymidine auxotrophy and CRISPR Device (CD). a** Experimental timeline. Specific-pathogen-free BALB/c mice were treated for 7 days with ciprofloxacin and metronidazole. *B. thetaiotaomicron* strains were administered by oral gavage on Day 0 or Day 1. The thymidine-auxotrophic *B. thetaiotaomicron* strains with ER and SpCas9 have either the sgRNA1 sequence or a non-targeting control (NT). **b** Outputs of NanoLuc in feces of mice (gavaged with *B. thetaiotaomicron* strains as indicated) measured until day 10. Each dot represents a biological replicate, and lines are the values of the mean calculated from the four replicates (Dunnet's T3 multiple comparisons test with Welch's one-way analysis of variance on log-transformed data at day10, ****$P < 0.0001$). **c** Viable *B. thetaiotaomicron* when fecal suspensions from mice gavaged with indicated strains without *thyA*⁺ (ER⁻/CD⁻) strain were plated on selective agar plates. Black bars are the values of the mean of four biological replicates. The average detection limits are $6.0 \times 10^{-2}$ CFU/mg Stool for *ΔthyA* (ER⁺/CD⁺) and $7.1 \times 10^{-2}$ CFU/mg Stool for *ΔthyA* (ER⁺/CD^NT), respectively. **d** Experimental timeline to test the function of CD in vivo. Germ-free C57BL/6 mice were gavaged with the *E. coli* S17-1 λ pir with the *thyA* gene on a mobilizable plasmid on Day 0, and then with *B. thetaiotaomicron* strains bearing ER and CD on Day 3. The *B. thetaiotaomicron* strains have either the sgRNA1 sequence or a non-targeting control (NT) in the sequence region of CD. **e** Viable *B. thetaiotaomicron* and *E. coli* when fecal pellets collected from mice gavaged with indicated strains on Day 7 and Day 14 were plated on selective agar plates and incubated in aerobic or anaerobic condition. Black bars indicate the values of the mean of four biological replicates (*n* = 4). The average detection limit is $1.1 \times 10^1$ CFU/mg Stool for *ΔthyA* (ER⁺/CD⁺). N.D⁻ represents no detection. Source data are provided as a Source Data file.

enumerated by plating the fecal homogenate on selective TYG agar plates without thymidine (for cells escaping from thymidine auxotrophy) and TYG agar plates with thymidine (for total viable cells) (Fig. 7a). Unexpectedly, some background growth appeared on these agar plates, but the colony morphology of *B. thetaiotaomicron* differed from that of the background. Therefore, we counted the colonies of the engineered strain by enumerating NanoLuc-positive colonies. Colonies escaping from thymidine auxotrophy were not detected in the fecal pellets (Fig. 7c). The average detection limits of the frequency of escape were $5.2 \times 10^{-7}$ for *ΔthyA* (ER⁺/CD⁺) and $1.1 \times 10^{-7}$ for *ΔthyA* (ER⁺/CD^NT), indicating that the thymidine auxotrophy in the engineered *B. thetaiotaomicron* strains was highly robust.

However, the number of viable thymidine-auxotrophic strains had not decreased after 7 days of the storage of solid feces at ambient temperature in the aerobic condition (Supplementary Fig. 10). This result is partially concordant with observations of *thyA*-deficient *B. ovatus* in the stools of mice[20], in which thymineless death was repressed in the presence of oxygen. Given that the CD accelerated the death rate in vitro, we expected that our containment system would cause the engineered *Bacteroides* strain to decline, even in feces in the presence of oxygen, more rapidly than noted by ref. [20]. The genetically engineered thymidine-auxotrophic strains could have been almost completely dormant without any replication of DNA or transcription of RNA in solid feces. Another possibility is the availability of substances such as nutrients that protected the thymidine-auxotrophic strains of *Bacteroides*.

To evaluate the genetic stability of the ER and CD, we isolated four colonies of the *ΔthyA* (ER⁺/CD⁺) strain from the feces of mice at day 10 and compared the isolates before and after inoculation. There was no significant difference in the luminescence values among the isolates, including controls before inoculation (Supplementary Fig. 11a). When the isolates were conjugated with *E. coli* carrying plasmids having the *thyA* gene, no viable *Bacteroides* cells were detected among the isolates derived from the feces. Thus, the CD remarkably inhibited the HGT of the *thyA* gene (Supplementary Fig. 11b). These results reveal that the ER and CD were stable for at least 10 days in the mouse gastrointestinal tract.

Finally, we validated whether the CD functions to prevent the disruption of thymidine auxotrophy by HGT of the *thyA* gene in vivo. Germ-free (GF) mice were first colonized with *E. coli* containing a mobilizable plasmid bearing the *thyA* gene. Subsequently, thymidine-auxotrophic *B. thetaiotaomicron* strains either with the ER and CD [*ΔthyA* (ER⁺/CD⁺)] or with the ER and non-targeting CD [*ΔthyA* (ER⁺/CD^NT)] were administered to assess whether there would be plasmid-borne transfer of the *thyA* gene from *E. coli* to *B. thetaiotaomicron* in the mouse gut (Fig. 7d). As expected, we observed colonies of the *ΔthyA* (ER⁺/CD^NT) strain on TYG agar plates both with and without thymidine when the stools of mice collected on day 7 were plated (Fig. 7e). In contrast, no colony of the *ΔthyA* (ER⁺/CD⁺) strain was detected on the TYG agar plates without thymidine, showing that the thymidine auxotrophy persisted. This result revealed that the CD had

prevented disruption of thymidine auxotrophy in vivo. However, the preventive effect was lost by day 14 (11 days after administration of *Bacteroides* strains), as the thymidine-auxotrophic *ΔthyA* (ER⁺/CD⁺) strain was replaced by a *B. thetaiotaomicron* strain without thymidine auxotrophy. We isolated twelve colonies of the non-auxotrophic strain and performed whole genome sequencing to clarify the mechanism of escape of the thymidine auxotrophy. The sequencing analysis showed that all twelve isolates lacked transgenes and had the *thyA* gene (BioSample accession number: SAMN37130011-SAMN37130022), whereas the ancestral strain contained the transgenes but not the *thyA* gene (BioSample accession number: SAMN37130023). We assume that a double crossover had occurred between the regions flanking the transgenes on the genome and homologous regions flanking the *thyA* gene on the plasmid. Both of the genome and the plasmid have the homologous sequences at the upstream (5′-*thyA*) and downstream (*thyA*−3′) of the *thyA* gene, as shown in Supplementary Fig. 2a and Fig. 4a. The transconjugant likely appeared via conjugation between *E. coli* and *B. thetaiotaomicron* after day 7 and the transconjugant with *thyA* grew and outcompeted the thymidine-auxotrophic *ΔthyA* (ER⁺/CD⁺) strain. It is noteworthy that the transgenes we introduced into the WT *B. thetaiotaomicron* were removed upon loss of the CD and acquisition of *thyA*. Therefore, the genetically engineered strain of *B. thetaiotaomicron* with the transgenes was still biocontained even after the CD failed to target the *thyA* gene, as the strain reverted back to the wild-type genotype in the *thyA* locus.

## Discussion

In this study, we describe an auxotrophic biological containment system that combines an ER and a CD. We demonstrate that this system is tunable for the expression of the GOI, that the *B. thetaiotaomicron* host strain does not acquire the *thyA* gene by HGT, and that the dissemination of transgenes is inhibited in this system.

It has been reported that an ER, which was established on plasmids in *E. coli*, could be applied to regulate levels of gene expression and create synthetic ribocomputing devices[43,44]. We expand this technology to a prevalent human commensal, *B. thetaiotaomicron*, by stably introducing the ER onto its genome. In *B. thetaiotaomicron*, the strength of gene expression is not largely affected by the level of RBS complementarity to the 16 S rRNA. This methodology could be applied to other human commensals that, like *Bacteroides*, rely on the strength of the S1 protein binding site for translation. The ER can also be applied as a safeguard when the taRNA and CR are designed for each individual GOI, by having the secondary structures of the crRNA include coding sequences. We created the taRNAs and CRs for the *nanoluc* gene as a model and showed the functionality of the tunable switch in *B. thetaiotaomicron*. The production of NanoLuc was switched on only when its gene was present in the cell with the CR and the corresponding taRNA. Moreover, by taking into consideration the MFE of the secondary structure when designing the taRNAs, an expression range of approximately 10-fold was obtained (Fig. 2f). A larger range of expression in the ON state and more strict suppression in the OFF state

would, however, be required for practical applications. The ER system is likely to be responsive to variations in temperature, considering the MFE. When the genetically engineered *B. thetaiotaomicron* comes out of the body, the production of the protein would not be expected to increase at ambient temperature, because there would be less translation of the GOI under the control of the ER. A deep-learning approach has already been developed to optimize the structure of ERs[45]. A more sophisticated regulatory system could be created by combining the use of the ER as a safeguard and improving the method.

It has been demonstrated that auxotrophy, created by eliminating essential genes, is effective for biological containment[11,12]. Nevertheless, it is possible that the containment system could be breached if the auxotroph were to acquire the essential genes via HGT from natural microorganisms[20]. We have constructed genetically modified thymidine-auxotrophic strains of *B. thetaiotaomicron* and quantified the loss of viable cells in the absence of thymidine. When the cells were cultured for up to 11 days, the CFU/mL of the $P_{cepA}$-sgRNA1 strain decreased by $10^7–10^8$. Given that the limit recommended by the National Institutes of Health[12] for the escape of engineered microbes or engineered DNA transmission is fewer than 1 cell per $10^8$ cells, the performance of this auxotrophic system falls just slightly short of that criterion. In particular, the death rate observed in the aerobic state was slower than that observed in the anaerobic state. The auxotrophic strain survived in the feces of mice in the presence of oxygen. Therefore, for clinical application, our system would require optimization. Essential genes or chassis strains could be chosen so as to create more suitable engineered auxotrophic microorganisms for employing the ER and CD in an uncontrolled environment. Nevertheless, the CD with a promoter of appropriate strength could help to increase the bacterial death rate in the absence of thymidine whereas the engineered microorganisms would grow in the presence of thymidine.

The function of the CD was verified by assessing the decrease in conjugation efficiency through filter mating. It was demonstrated that the CD specifically recognizes a sequence identified by a sgRNA and makes a double-strand break in the sequence. This characteristic would allow us to precisely target an unfavorable gene that could potentially destroy a biological containment system. However, preventing HGT by the CD could potentially be a drawback, because theoretically, numerous variants of the target gene are present in the environment. A preferable candidate as a target sequence would be one chosen from well-conserved sequences in a gene, such as *thyA* and *dapA*, whose deficiency causes auxotrophy, leading to rapid death unless an essential nutrient is supplied[46,47]. Furthermore, introducing multiple sgRNAs would help to cover the wider range of potential variants[34].

The CD suppressed the frequency of acquisition of the *thyA* gene and transgenes by 1 to 2 orders of magnitude, which seems to be adequate to prevent the acquisition of those genes, considering the conjugation efficiency of *B. ovatus* and the target frequency described above. The conjugation efficiency of this species for the *thyA* gene on its genome has been reported as $0.9–4.2 \times 10^{-7}$ while the recommended efficiency is below $1 \times 10^{-8}$ [20]. If the CD is utilized for gene modification, its efficiency would satisfy this criterion. The CD could be optimized by adjusting the strength of the promoters, as shown in Fig. 5b, though the CD does affect cell growth, so there is a tradeoff. Potentially, the capability could also be improved by using Cas3 and the CasABCDE complex, which is analogous to Cas9 and enhances DNA degradation after the initial cleavage with 3′-to-5′ helicase and ssDNA exonuclease activities[34]. Transgenes are unlikely to be expressed by species not closely related to *Bacteroides* because of the specific regulation of gene expression[48–50], even if the transgenes have transferred. In choosing promoters, we considered both intraspecies HGT and HGT between closely related species. For the CD, we chose active promoters in interspecies populations of *Bacteroides*: $P_{cepA}$ or $P_{cfxA}$ for the transcription of sgRNA, and $P_{cfiA}$ for that of SpCas9[49–52]. If promoters that would function in a broader range of gut bacterial cells are needed

to effectively prevent the dissemination of transgenes, genomic analysis of gut microbiota could be used to obtain information on widely distributed promoters; in fact, some promoters have already been reported to function in a variety of gut bacterial cells[53]. Therefore, functionality can be flexibly tuned to adapt to particular conditions of use. When tested in the mouse gut, the CD prevented the disruption of thymidine auxotrophy for four days. However, the extended experiment showed that the CD was not flawless: there is a possibility that the CD would eventually fail to guard thymidine auxotrophy. The improvements described above would be useful for a more stringent auxotrophic system. In addition, the failure of the CD could be prevented by introducing transgenes at the other sites of *thyA* locus in a *thyA*-deficient strain. This genetic modification avoids loss of the CD which would be caused by genetic exchange between the transgenes and an exogenous *thyA* gene. Nevertheless, this containment system is highly reliable since the genetically engineered strains with both transgenes and *thyA* gene were not detected even when the thymidine auxotrophy did not persist.

The phenotypes introduced by genetic engineering via double crossover, such as the thymidine auxotrophy, the ER, and the CD, were highly robust in vivo as well as in vitro. Those phenotypes in the genetically engineered strains of *B. thetaiotaomicron* were maintained after 10 days of colonization in the mouse gut. The persistent gene expression of Nanoluc regulated by the ER was confirmed in vivo. The CD also efficiently prevented the disruption of thymidine auxotrophy in the gastrointestinal tract by inhibiting the HGT of *thyA* gene from the surrounding environment.

Enzymes synthesized by the human microbiota generate many kinds of metabolites that affect host physiology, such as short chain fatty acids and bile acids[54,55]. Genetically modified microorganisms bearing the related genes, as well as IL-10-producing *Lactococcus lactis* mentioned above, might offer a new therapeutic approach to treat diseases caused by an imbalance of these metabolites. For these organisms to be used effectively, for example, to avoid unintended protein overproduction by the bacteria or the spread of the introduced gene to other bacteria, the expression level of the desired proteins would have to be regulated. Therefore, the framework of this Cas9-assisted biological containment has great potential for the practical use of genetically modified microorganisms in biomedicine and other fields.

We have provided proof-of-concept that the Cas9-assisted thymidine auxotrophic biocontainment system is tunable and useful for simultaneously preventing the dissemination of transgenes and the escape from thymineless death, as well as controlling the level of gene expression. The ON/OFF ratio of the gene expression and the death rate in unfavorable conditions still require improvement to meet the standards for clinical application. One way to achieve a system that can be applied in clinical situations is to use other human commensal bacteria or essential genes, and to design the ER with deep learning. By addressing these challenges, future work can lead to the availability of useful genetically modified microorganisms.

## Methods

### Strains and culture conditions

*B. thetaiotaomicron* VPI-5482 (ATCC 29148) (GenBank: AE015928.1) was used for all evaluation of biological containment of genetically modified bacterial strains and genetic elements. This strain, which is sensitive to erythromycin but resistant to gentamicin, was grown in Brain Heart Infusion (Difco) media with 5 g/L yeast extract (Bacto) (BHIS), Trypticase-Yeast extract-Glucose (TYG) media or Defined Minimal Media (DMM), supplemented with 1 mg/L resazurin sodium salt (Alfa Aesar), 10 mg/L hemin (Sigma), 0.5 g/L cysteine hydrochloride (Sigma), and 1 mg/L Vitamin K3 (Sigma)[31]. TYG media contained: 10 g/L trypticase (BBL), 5 g/L yeast extract (Bacto), 1 g/L $Na_2CO_3$ (AMERSCO), 2 g/L glucose (Sigma), 80 mM potassium phosphate buffer (pH 7.3), 20 mg/L $MgSO_4{\cdot}7H_2O$ (Sigma), 400 mg/L $NaHCO_3$

(Sigma), 80 mg/L NaCl (MACRON), 8 mg/L $CaCl_2$ (Sigma), and 0.4 mg/L $FeSO_4 \cdot 7H_2O$ (Sigma). DMM media contained: 1 g/L $NH_4SO_4$ (Sigma), 1 g/L $Na_2CO_3$, 80 mM potassium phosphate buffer (pH 7.3), 900 mg/L NaCl (MACRON), 20 mg/L $CaCl_2$ (Sigma), 20 mg/L $MgCl_2 \cdot 6H_2O$ (VWR), 10 mg/L $MnCl_2 \cdot 4H_2O$ (Sigma), 10 mg/L $CoCl_2 \cdot 6H_2O$ (Sigma), 5 µg/L Vitamin B12 (Sigma), 4 mg/L $FeSO_4 \cdot 7H_2O$ (Sigma), and 5 g/L carbon source. Cultures were grown in an anaerobic chamber with an atmosphere of 2% $H_2$ balanced with $N_2$ and $CO_2$ at 37 °C. All media was pre-reduced overnight in an anaerobic atmosphere before inoculation. Antibiotics, thymidine, carbon sources, bile acids, 4-Chloro-DL-phenylalanine, and agarose were added as appropriate: erythromycin (Sigma, 25 µg/mL), gentamicin sulfate (TEKnova, 200 µg/mL), carbenicillin disodium salt (Gold Biotechnology, 5 µg/mL), tetracycline hydrochloride (IBI Scientific, 5 µg/mL), Polymyxin B sulfate (Merck Millipore, 10 µg/mL), thymidine (Alfa Aesar, 100 µg/mL), galactose (Sigma, 0.5% w/v), maltose (Sigma, 0.3% w/v), starch from potato (Sigma, 0.5% w/v), deoxycholic acid (Sigma, 62.5 µM), sodium choleate (Sigma, 0.5%) as a crude ox bile extract, 4-Chloro-DL-phenylalanine (Sigma, 2 mg/mL), and agarose (LabExpress, 15 g/L).

*E. coli* S17-1 λ pir[56] was utilized for the construction of plasmids and the transformation of *B. thetaiotaomicron* strains through conjugation. *E. coli* S17-1 λ pir was routinely grown in Luria Bertani (LB) broth (LabExpress) supplemented with 100 µg/mL carbenicillin (Gold Biotechnology) for plasmid selection.

## Plasmid construction

All plasmids were constructed with standard cloning procedures. Briefly, DNA fragments were generated by PCR or purchased from Integrated DNA Technologies, Inc. Fragments were assembled by Gibson Assembly Cloning Method into plasmids[57]. Plasmid sequences were confirmed by Sanger sequencing. *E. coli* S17-1 λ pir was transformed with the constructed plasmids by electroporation to propagate the plasmids and for transformation of *B. thetaiotaomicron* through conjugation. Genetic parts, plasmids, and primers used in this study are shown in Supplementary Tables 1–6. The representative plasmids are depicted in Supplementary Fig. 1: pNH9188 was used for the first recombination and pNH9196 was used for the second recombination to construct the genetically engineered strains of *B. thetaiotaomicron*, which were evaluated in vitro and in vivo.

## Strain construction

Thymidine-auxotrophic genetically modified strains were generated through recombination of plasmids bearing the ER and CD (Supplementary Fig. 1) into the genome of *B. thetaiotaomicron*. The wild-type *B. thetaiotaomicron* VPI-5482 strain was transformed with the plasmids to build genetically modified *thyA*-deficient strains through double crossovers upstream and downstream of the *thyA* locus (Supplementary Fig. 2)[58]. The taRNA sequences, SpCas9 gene, CR, and NanoLuc gene were integrated into the genome of the *B. thetaiotaomicron* by the first recombination, in which the *thyA* gene on the genome was exchanged with transgenes. sgRNA to target the *thyA* gene was integrated by the second recombination (Supplementary Fig. 2a). The plasmids bearing those sequences were used to transform WT *B. thetaiotaomicron* through conjugation with *E. coli* S17-1 λ pir.

Overnight cultures of the WT strain were added to *E. coli* S17-1 λ pir with the plasmids and the mating mixtures were pelleted, resuspended in BHIS medium, spotted onto BHIS agar plates and incubated upright at 37 °C aerobically. After overnight incubation, cells were collected by scraping, resuspended, and plated on BHIS agar plates with gentamicin and erythromycin as selective markers and incubated anaerobically for 2 days at 37 °C. Resultant colonies were re-isolated on the BHIS agar plates with gentamicin and erythromycin.

To obtain the *thyA*-locus-exchanged mutant, each colony was anaerobically cultured for 1 day in 3 mL of BHIS with thymidine without antibiotics. Bacterial cells were collected by centrifugation, washed, and resuspended in PBS, and the appropriate dilution was plated onto DMM agar plates (0.5% glucose) supplemented with p-Chlorophenylalanine. The colonies collected from the DMM plates after 3 days of cultivation at 37 °C were streaked on TYG and TYG with thymidine plates to check their sensitivity to thymidine. After a 48-h cultivation, colonies on TYG with thymidine plates were restreaked on TYG with thymidine and TYG with thymidine and erythromycin plates to check their sensitivity to erythromycin. Strains sensitive to thymidine and erythromycin were taken as *thyA*-locus-exchanged mutants.

The genetic exchange was confirmed by PCR using primers which bound to the regions flanking *thyA* or to the sgRNA region. DNAs were purified from both thymidine-auxotrophic and erythromycin-sensitive colonies using the DNeasy Blood & Tissue Kits (Qiagen). After the DNA concentration was determined, PCR was performed using primers encompassing the exchanged site (Supplementary Fig. 2a; Supplementary Table 6). The amplicon sizes were compared with those from the parental strain by agarose gel electrophoresis to confirm that the expected exchange occurred. PCR products of 2885 bp, including *thyA*, were detected in the WT strain. On the other hand, about 8000 bp of the amplicon that was bearing transgenes were detected after the double crossovers were introduced, indicating that the *thyA* gene on the chromosome had been exchanged through the recombination (Supplementary Fig. 2b).

The second recombination, i.e., that between plasmids bearing sgRNAs downstream of promoter $P_{cepA}$ or $P_{cfxA}$ and the chromosome of genetically modified *B. thetaiotaomicron*, was performed to integrate those sequences. The recombination process was the same as the first recombination except that thymidine was added to all of the media and agar plates. PCR was performed using primers encompassing the exchanged site and the purified DNAs as template. The presence of amplicons was checked by agarose gel electrophoresis to confirm that the expected integration had occurred. After the second recombination, the amplicons were detected at the length of about 6000 bp in strains bearing sgRNAs on the genome, whereas no amplicon was found in the parental strain (Supplementary Fig. 2c).

## Design of engineered riboregulator (ER)

The ERs composed of crRNA and taRNA were designed with RNAfold WebServer, which is used to predict nucleic acid structures[38]. Sequences of 99 nucleotides were simulated for crRNA; the 99-nucleotide sequences were composed of the RBS, CR, and part of the NanoLuc coding sequence, which contains the region considered to contribute to the mRNA folding effect on translation[31,59]. Likewise, structures of 99-nucleotide sequences of taRNA were predicted. The MFE of RNA folding was predicted for each variant using default parameters.

Structures of the crRNA/taRNA complex were also analyzed with NUPACK web server[60], used to predict the MFE of mRNA structures including rpiL* and part of the NanoLuc coding sequence in the previous study. NUPACK web server can be applied to multiple complex RNA structures whereas RNAfold WebServer analyzes a single RNA structure[38]. The MFE was calculated using default parameters other than concentrations of the crRNA and taRNA, and maximum complex size. We estimated the concentrations of the crRNA and taRNA to be 100 nM and the maximum complex size to be 2[61].

## NanoLuc luciferase assay in vitro

Overnight cultures in TYG medium with thymidine were diluted 1:100 in fresh medium. The cultures were grown anaerobically at 37 °C to an optical density ($OD_{600}$) of 0.4–0.8, and the cultures were harvested by centrifugation at $4500 \times g$ for 10 min, washed once, and resuspended with phosphate buffered saline (PBS). NanoLuc Reaction Buffer (Promega) and cell suspensions were mixed at equal volumes. The luciferase activities were measured with an integration time of 1 s at a gain setting of 100 in BioTek Synergy H1 Hybrid Reader 6 min after mixing. The luminescent values were normalized to $OD_{600}$ of 300 µL of cell

suspensions in PBS. Fold changes were calculated on the basis of the value with no CR (rpiL*) and non-sense taRNA, or the value with crRCN and non-sense taRNA.

## Growth curve of genetically modified thymidine-auxotrophic *B. thetaiotaomicron*

Overnight cultures grown in TYG medium or DMM medium (0.5% glucose) with thymidine were harvested by centrifugation at $3200 \times g$ for 10 min, washed with PBS twice, and resuspended in fresh TYG medium without thymidine or DMM medium without either thymidine or glucose. The cell suspensions were diluted 1:100 in the fresh TYG medium or DMM medium with or without thymidine. For cell growth in various stress conditions, fresh DMM medium with or without thymidine was used to dilute the cell suspensions; this medium included appropriate carbon sources or bile acids, as well as fresh TYG medium with or without thymidine. The cultures were grown anaerobically at 37 °C for 24 h. Samples (300 μL) were withdrawn from the cultures for 10 h and at 24 h to monitor growth ($OD_{600}$) in a BioTek Synergy H1 Hybrid Reader.

## Viability test of genetically modified thymidine-auxotrophic *B. thetaiotaomicron* in vitro

Overnight cultures grown in TYG or DMM medium (0.5% glucose) with thymidine were centrifuged and washed twice in PBS, and cells were resuspended in fresh TYG without thymidine or in fresh DMM medium without either thymidine and glucose. Cell suspensions were diluted in TYG, TYG medium with bile acids, or DMM media with different carbon sources with or without thymidine, so that the optical density of the cell suspensions at 600 nm ($OD_{600}$) was 0.2. For carbon source starvation, bacterial cells grown in DMM with 0.5% glucose were pelleted ($OD_{600} = 0.4$), washed three times in PBS, resuspended in DMM without glucose in the presence or absence of thymidine, and incubated for up to 100 h. Cultures were incubated anaerobically at 37 °C or aerobically at ambient temperature. Inocula were withdrawn, and colony forming units (CFU) were measured by plating serial dilutions on BHIS with thymidine and gentamicin in an anaerobic chamber. After incubation at 37 °C for more than 4 days, colonies were counted manually and corrected for the dilution factor to obtain CFU/mL.

## Assay for inhibition of the Horizontal Gene Transfer (HGT) of *thyA* gene in vitro

Overnight cultures of genetically modified thymidine-auxotrophic *B. thetaiotaomicron* strains were diluted 1:100 in fresh TYG medium with thymidine, and *E. coli* S17-1 λ pir strains bearing plasmids with the *thyA* gene and an erythromycin resistance gene were diluted 1:100 in fresh in LB medium with carbenicillin. The cultures were grown anaerobically for *B. thetaiotaomicron* and aerobically for *E. coli* at 37 °C to an optical density of 0.4–0.9 at 600 nm, then harvested by centrifugation at $3200 \times g$ for 10 min, washed in PBS twice, and resuspended in the fresh TYG medium with or without thymidine. The optical density of the cell suspension was adjusted to 16 with the medium, and *B. thetaiotaomicron* and *E. coli* were mixed at equal volumes. Then, 50 to 100 μL of the mixtures were transferred onto a 0.45 μm filter disc placed on TYG agar plates, with or without thymidine, and incubated anaerobically at 37 °C overnight. To evaluate conjugation efficiency, cells were washed off the filter and resuspended in BHIS medium with gentamicin and thymidine or in TYG medium with gentamicin. The cell suspension was plated on BHIS agar plates with gentamicin, thymidine, and erythromycin or on TYG agar plates with gentamicin. Total CFU/mL of viable cells after conjugation were determined by plating the cell suspension on BHIS agar plates containing gentamicin and thymidine or TYG agar plates containing gentamicin and thymidine. The colonies were counted manually 4 days after culturing.

DNA was purified from transconjugants grown on BHIS agar plates with gentamicin, thymidine, and erythromycin after conjugation, using the DNeasy Blood & Tissue Kits (Qiagen). PCR was performed with

primers *thyA*-F and *thyA*-R to detect the *thyA* gene. Primers named qPCR-EmR-F and qPCR-EmR-R were used for the EmR gene (Supplementary Table 6). Whole genome sequencing was also performed with the purified DNA to check for the integration of the plasmid bearing the intact *thyA* gene.

## Assay for prevention of the Horizontal Gene transfer (HGT) of transgenes

Overnight cultures of WT *B. thetaiotaomicron* strains were diluted 1:100 in fresh TYG medium with thymidine, and *E. coli* S17-1 λ pir strains bearing plasmids with transgenes, such as taRNA, sgRNA, SpCas9, CR and NanoLuc gene, and an erythromycin resistance gene were diluted 1:100 in fresh LB medium with carbenicillin. The cultures were grown anaerobically for *B. thetaiotaomicron* and aerobically for *E. coli* at 37 °C to an optical density of 0.4–0.9 at 600 nm. Cultures were harvested by centrifugation at $3200 \times g$ for 10 min, washed twice in PBS, and resuspended in fresh TYG medium without thymidine. The optical density of the cell suspension was adjusted to 16 with the medium, and *B. thetaiotaomicron* and *E. coli* were mixed at equal volumes. Then, 50 μL of the mixtures were transferred onto a 0.45 μm filter disc placed on TYG agar plates without thymidine. Plates were incubated anaerobically at 37 °C overnight. To evaluate the number of cells which had acquired plasmids with the transgenes and the erythromycin resistance gene, cells were washed off the filter and resuspended in the BHIS medium containing gentamicin, thymidine, and erythromycin. The cell suspension was plated on the BHIS agar plates containing the same components. The colonies were counted manually 4 days after culturing.

DNA was purified from transconjugants on BHIS agar plates with gentamicin, thymidine, and erythromycin after conjugation, using the DNeasy Blood & Tissue Kits (Qiagen). PCR was performed with primers Seq-*thyA*-F and mmD663 to detect the region including the CD, and qPCR-EmR-F and qPCR-EmR-F for the erythromycin resistance gene (Supplementary Table 6). Whole genome sequencing was also performed with the purified DNA to check the integration of the plasmid bearing the transgenes and the erythromycin resistance gene.

## Whole genome sequencing and assembly

Total DNA of erythromycin-resistant colonies was purified using the DNeasy Blood & Tissue Kits (Qiagen). Genomic DNA library preparation and sequencing were performed at the SeqCenter (Pittsburgh, PA) by using Illumina NextSeq 2000 and NovaSeq 6000 technology. Sample libraries were prepared using the Illumina DNA Prep kit and IDT 10 bp unique dual indices (UDI). Demultiplexing, quality control and adapter trimming were performed with BCL Convert (Version 4.1.5). To analyze genetic alterations by HGT, paired-end sequencing reads ($2 \times 151$ bp) were assembled by SPAdes (Version 3.15.2)[62]. The sequencing reads ($2 \times 151$ bp) were also mapped to the plasmid bearing the *thyA* gene (shown in Fig. 4a) and the plasmid with transgenes (shown in Fig. 5a), using Bowtie 2 (Version 2.4.5)[63]. Raw sequencing data and assembled genomes have been deposited at DDBJ/ENA/GenBank under the accession number: PRJNA754595.

## Evolutionary stability test

The stability of the CD and ER during cell growth was verified by culturing the genetically modified thymidine-auxotrophic *B. thetaiotaomicron* strain with those functions for up to 21 days. The expression of the NanoLuc gene and the ability to block the HGT of the *thyA* gene were checked every 7 days. Overnight cultures in TYG medium with thymidine were diluted 1:100000 in fresh medium and grown anaerobically at 37 °C for 24 h, then grown continuously with dilution into fresh TYG media with thymidine to $10^{-5}$ of optical density at 600 nm every $24 \pm 2.5$ h[34]. The functioning of the CD and ER was evaluated on samples withdrawn from the passaged culture in accordance with the protocols of the NanoLuc luciferase assay and the assay for frequency

of HGT of the *thyA* gene mentioned above. Samples were grown in continuous culture and measured periodically, as described.

DNA was purified from transconjugants grown on BHIS agar plates with gentamicin, thymidine, and erythromycin after conjugation, using the DNeasy Blood & Tissue Kits (Qiagen). PCR was performed with primers Seq-*thyA*-F and mmD663 to amplify the region including the CD, and qPCR-EmR-F and qPCR-EmR-F to detect the erythromycin resistance gene (Supplementary Table 6). Sanger sequencing of the portion from the $P_{cepA}$-sgRNA1 to the terminator downstream of SpCas9 was performed with the purified amplicons to check if there were any mutations in the CD region. Whole genome sequencing was also performed with the purified DNA to check the integration of the plasmid bearing the intact *thyA* gene.

## Animals

The protocols of SPF animal experiments were approved by the MIT Committee on Animal Care. Female BALB/cJ mice (8 weeks) were purchased from The Jackson Laboratory, and four mice in each experimental group were housed together with a 12-h light/ dark cycle at standard room temperature of 20–22 °C at a humidity of 30–70%, and handled in non-sterile conditions. Mice were fed an irradiated mouse diet (ProLab IsoPro RMH3000, LabDiet) and non-sterile water before treatment with antibiotics. Prior to bacterial administration, mice were transferred to clean cages and then gavaged with metronidazole (100 mg/kg) in sterile water every day for 7 days. Over the course of the treatment, they were provided sterile filtered water containing ciprofloxacin hydrochloride (0.625 g/L). Animals were transferred to clean cages and provided fresh medicated water on the third or fourth day of the antibiotic regimen. Two days after the cessation of antibiotic treatment, they were transferred to clean cages and inoculated with *B. thetaiotaomicron* strains (~4 × 10$^8$ CFU/mouse) by oral administration. Fecal pellets were collected for the NanoLuc luciferase assay and the viability test of the bacterial strains in mice feces.

All gnotobiotic experiments were performed in accordance with the Guide for the Care and Use of Laboratory Animals (8th ed.) and were approved by the Institutional Animal Care and Use Committee of the University of Chicago. Female germ-free C57BL/6 mice were bred and housed in Trexler-style flexible film isolators (Class Biologically Clean) within polycarbonate mouse cages. Mice were weaned at 21 days of age and were fed an autoclaved, plant-based diet (JL Rat and Mouse/Auto 6 F 5K67, LabDiet). All mice were housed in cages with a 12-h light/ dark cycle at standard room temperature of 20–24 °C at a humidity of 30–70%. 8–12 week-old mice were inoculated with the *E. coli* S17-1 λ pir strain bearing plasmids with the *thyA* gene (~10$^8$ CFU/mouse) and transferred into bioexclusion cages (IsoCage P, Techniplast). After 3 days, genetically engineered *B. thetaiotaomicron* strains (~10$^8$ CFU/ mouse) were admistered by oral gavage. Fecal pellets were collected to assay the inhibition of the HGT of the *thyA* gene in the mouse gut.

## NanoLuc luciferase assay in vivo

Fecal pellets were homogenized in PBS with an autoclaved single 5 mm stainless steel bead (Qiagen) using TissueLyser II (Qiagen) at 25 Hz for 2 mins. The concentration of feces was adjusted to less than 27 mg/mL. The fecal suspensions were centrifuged at $500 \times g$ for 30 s to precipitate debris. The supernatants were diluted 10-fold in PBS. NanoLuc Reaction Buffer (Promega) and supernatants were mixed at equal volumes, and luciferase activities were measured with an integration time of 1 s at a gain setting of 100 in BioTek Synergy H1 Hybrid Reader 6 mins after mixing. The luminescent values were normalized to the weight of the stools.

## Viability test of genetically modified thymidine-auxotrophic *B. thetaiotaomicron* in vivo

Fecal pellets were collected and stored in the presence of oxygen at ambient temperature prior to evaluation. Pellets were homogenized in

PBS with a single autoclaved 5 mm stainless steel bead (Qiagen) using TissueLyser II (Qiagen) at 25 Hz for 2 min. The fecal suspensions were centrifuged at $500 \times g$ for 30 s to precipitate debris.

Supernatants were serially diluted under anaerobic conditions. The dilutions were plated on TYG agar plates either with erythromycin, gentamicin, carbenicillin, and polymyxin B, or with erythromycin, gentamicin, carbenicillin, polymyxin B, and thymidine. Luminescent colonies were counted manually when the NanoLuc Reaction Buffer (Promega) was spotted on the colonies, four days after culturing. The CFU values were normalized to the weight of the stools.

## Assay for inhibition of the Horizontal Gene Transfer (HGT) of *thyA* gene in vivo

Fecal pellets were homogenized in 1 mL of PBS with in a PowerLyzer 24 homogenizer (Qiagen) for 2 min at 2000 rpm, followed by a quick 30 s spin at 300 g to precipitate debris. Supernatants were serially diluted under anaerobic conditions. The dilutions were plated on LB agar plates to measure CFU/mg stool of the *E. coli* strain, on TYG agar plates with gentamicin for the *B. thetaiotaomicron* strains which overcame thymidine auxotrophy, or on TYG agar plates with gentamicin and thymidine for total *B. thetaiotaomicron* strains including thymidine-auxotrophic and non-auxotrophic strains. Colonies were counted manually one day after culturing aerobically on the LB agar plates or two days after culturing anaerobically on the TYG agar plates. The CFU values were normalized to the weight of the stools.

DNA was purified from colonies grown on TYG agar plates with gentamicin, using the DNeasy Blood & Tissue Kits (Qiagen). Whole genome sequencing was performed with the purified DNA to check for the integration of the plasmid bearing the *thyA* gene in the *E. coli* strain.

## Statistics

All data were analyzed using GraphPad Prism 9 (GraphPad Software).

## Reporting summary

Further information on research design is available in the Nature Portfolio Reporting Summary linked to this article.

## Data availability

Data supporting this study are presented in the main text and Supplementary information, and source data are provided with this paper. Raw sequencing data and assembled genomes of whole-genome sequencing of *Bacteroides thetaiotaomicron* strains have been deposited at DDBJ/ENA/GenBank under the accession number: PRJNA754595. Plasmids used in this study are available from the corresponding author upon request.

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

## Acknowledgements

We would like to thank all lab members in the Synthetic Biology Group in MIT's Synthetic Biology Center for their great help, especially, Dr. Ky Lowenhaupt, Dr. Isaak E. Mueller, Dr. Giyoung Jung, Dr. Tzu-Chieh Tang, Dr. Kevin M. Yehl and Dr. Maria Eugenia Inda. We thank Karen Pepper for editing the manuscript. We thank Dr. Xiaoqiong Gu for bioinformatics analysis. We also thank the staff at the Koch Institute Swanson Bio-technology Center and the Division of Comparative Medicine at MIT for their assistance for animal experiments. We thank the Duchoissois Family Institute for providing germ-free mice. This study was supported by JSR Corporation in Japan. This work was supported by the National Science Foundation (NSF-CCF-1521925 to T.K.L.), the National Institutes of Health (NIH-5-U01-CA2550554-02 and NIH-50000655-5500001351 to T.K.L., NIGMS R35GM147478 to M.M., and IMSD Program 5R25GM109439-07 and Molecular and Cellular Biology training pro-gram T32 GM007183 to J.F.-S.), and the Arnold and Mabel Beckman Foundation through the Beckman Young Investigator Program (M.M.).

## Author contributions

N.H., Y.L., M.M. and T.K.L. conceived and designed this study. N.H., Y.L. and J.F.-S. performed the experiments. N.H., Y.L., J.F.-S., M.M. and T.K.L. analyzed data, discussed the results, and wrote the manuscript.

## Competing interests

T.K.L. is a co-founder of Senti Biosciences, Synlogic, Engine Biosciences, Tango Therapeutics, Corvium, BiomX, Eligo Biosciences, Bota.Bio, and Avendesora. T.K.L. also holds financial interests in nest.bio, Ampliphi, IndieBio, MedicusTek, Quark Biosciences, Personal Genomics, Thryve, Lexent Bio, MitoLab, Vulcan, Serotiny, and Avendesora. N.H. is an employee of JSR Corporation. Other authors declare no competing interests.
