## [Peer Review File · Nature Communications]

Reviewers' Comments:

Reviewer #1:

Remarks to the Author:

In this study, the authors have developed a Cas9-based biological containment system with an engineered riboregulator for the controlled gene expression in *Bacteroides thetaiotaomicron*. Biological containment is an essential safety feature required in engineered microbes. The authors have shown that the targeting of *thyA* using CRISPR-Cas9 can prevent the engineered bacteria from escaping containment, as the escape is generally mediated by acquiring *thyA* gene via horizontal gene transfer. In addition, a split engineered riboregulator was developed with a trans- and a cis-acting elements that prevent the expression of the gene of interest in case the partial cassette is transferred to another host. The authors validated the functioning of the systems in in-vitro and in-vivo studies.

Although the authors show the functionality of the engineered riboregulator and the Cas9-containment systems, there are some drawbacks to the study. The riboregulator described here is not novel as similar RNA-based regulators have been reported before. It is not immediately clear if the riboregulator reported here has any advantage compared to those previously reported. In addition, the biological containment system designed to counter horizontal gene transfer was not sufficiently proven to be effective in-vivo as no gene transfer was observed even in the control strain in this study. I suggest that the following comments be addressed during revision.

Major comments

1. The engineered riboregulator is very similar to those described in previous publications (see Isaacs, F., Dwyer, D., Ding, C. et al. *Nat Biotechnol* 22, 841–847 (2004)). Please discuss and highlight the differences and advantages of the riboregulator described in this study compared to those already reported.
2. The temperature is likely to influence the activity of the riboregulator as the structure of the folded trans- and cis- RNA elements will change. Please discuss how this system behaves at temperatures other than physiological temperature.
3. Fig 3f shows that the presence of the oxygen inhibits thymineless death of the bacteria due to the decrease in cell growth. Please check if other stressful conditions, such as limited nutrient availability, are likely to have the same effect on the auxotrophy in the bacteria.
4. In Fig 6, the evolutionary stability of ER and CD should be checked in conditions that better represent the physiological conditions to accurately predict how these genetic elements will behave once the bacteria colonize the gastrointestinal tract.
5. The in-vivo study did not prove that CD prevents the escape of the bacteria from biological containment as no horizontal gene transfer was observed in the study (as depicted by the low frequency of the escape of the strain with the non-targeted sgRNA). As the main crux of this study is the use of CD for biological containment, it should be demonstrated in an in-vivo model. Perhaps the authors can make use of a bacterial strain with a plasmid that is more readily able to undergo horizontal gene transfer into the *B. thetaiotaomicron* with ER and CD.

Minor comments

1. Please provide a reference for lines 154-157 on page 7.
2. Figure 3b, please mention the amount of thymidine supplemented in the media.
3. For the filter mating experiment in Fig 4b and 5b, the selection media was BHIS with erythromycin. Please check if this is correct as this media will not select against the *E. coli* donor.
4. Fig 7a should be redrawn. The figure currently suggests that in all groups, the WT strain was administered on Day 0.
5. As significant toxicity is observed due to CD, perhaps the authors can include in the discussion how this can be mitigated by using repressible/inducible systems.
6. Please discuss the likelihood and the consequence of the horizontal gene transfer of the partial gene circuit, such as only the gene of interest.
7. The authors have shown that CD fails to prevent horizontal gene transfer in case the target gene is mutated. As such, please discuss how the CD will be able to prevent the escape from biological containment as there are potentially numerous variants of the target gene in the environment.

Reviewer #2:

Remarks to the Author:

The manuscript by Hayashi et al is a well written and well presented study on the engineering of a biocontainment system for engineered DNA in a *Bacteroides* strain capable of colonising the human gut microbiome. Previously this strain was put forward by this research group and others as a promising chassis organism for engineering in the human gut microbiome. A challenge for putting engineered bacteria in the human gut is biocontainment and a genetic engineering solution to that is proposed and tested here. The proposed approach centres around thymidineless death and thymidine auxotrophy - a long established method to have bacteria only able to survive when thymidine is provided. The authors here upgrade this simple system with some clever tricks designed to prevent HGT and some aspects of mutation finding a way around the trophic containment. Simply put, they surround their engineered 'cargo' gene with a CRISPR device that cuts the *thyA* gene and flank that device with a pair of optimised riboregulator parts that control the expression of the cargo gene. If the CRISPR device is deleted, then its very likely that the riboregulator parts will be separated and so cargo gene expression will not work. The whole construct is added to the genome to replace the *thyA* gene. The authors run multiple tests of different designs, especially the riboregulator parts and guide RNA choices, and show the containment system working well in vitro and in vivo too in mice models. They only ever show it's performance with a single cargo gene (NanoLuc) but this is good enough as a representation. By the end of the manuscript the system is shown to be performing very well, but falls slightly below the goal set by the NIH. Anaerobic vs. Aerobic conditions, and the presence of small amounts of thymidine naturally in the gut seem to pose problems that stem not from the engineering done here, but in the thymidine trophic selection system in general. In the discussion the authors cover how improvements to their work could solve these minor issues.

I liked this work, and I think it is well described, innovative and timely. It is a good fit for the journal readership and I'm supportive of its publication without any major revisions needed. In fact, I could only find 3 minor points that I think need addressing:

1. Line 65 - xenobiology is put forward as a biocontainment system, but the authors only discuss XNA xenobiology which I think has never been practically applied in bacteria in application. I was surprised that genetic code 'xenobiology' was not mentioned here, where non-canonical amino acids (ncAAs) are used for biocontainment. George Church lab and Farren Isaacs lab both published remarkably good biocontainment results from using the recoded *E.coli* genome and having TAG codons placed in essential genes in a way which requires ncAAs to be present for those genes to fold and function. The best of these is citation #13 in the manuscript, and is cited in the auxotrophy section of the intro instead. I think this is a bit of a mix-up and I would instead consider this work fitting in better as part of the 'xenobiology' section. In fact, I would probably change the name to 'genetic firewall' rather than xenobiology and then the codon recoding that prevents HGT (e.g. by phages) in these organisms can also be covered - e.g. with Jason Chin's Syn61 *E.coli*.
2. Figure 2h legend - please say in the legend what the green and black bars represent
3. Line 335 - the promoter used presumably has to operate in a broad range of host cells for the system to work. Can the authors detail their approach to choosing a promoter that is predicted to function in a broad range of gut bacteria cells? Either here or in the discussion section.

Cas9-Assisted Biological Containment of a Genetically Engineered Human Commensal Bacterium and Genetic Elements

NCOMMS-21-44334

Point-By-Point Response to Reviewers' Comments

We thank both reviewers and the editor for their time reviewing our manuscript. We have responded to their concerns and described the changes we have made in response to each comment. We have conducted additional experiments and have added revised figures and tables to the manuscript. We appreciate both reviewers' inputs in helping us achieve this improved version of our manuscript.

Reviewer #1 (Remarks to the Author):

In this study, the authors have developed a Cas9-based biological containment system with an engineered riboregulator for the controlled gene expression in *Bacteroides thetaiotaomicron*. Biological containment is an essential safety feature required in engineered microbes. The authors have shown that the targeting of *thyA* using CRISPR-Cas9 can prevent the engineered bacteria from escaping containment, as the escape is generally mediated by acquiring *thyA* gene via horizontal gene transfer. In addition, a split engineered riboregulator was developed with a trans- and a cis-acting elements that prevent the expression of the gene of interest in case the partial cassette is transferred to another host. The authors validated the functioning of the systems in in-vitro and in-vivo studies.

Although the authors show the functionality of the engineered riboregulator and the Cas9- containment systems, there are some drawbacks to the study. The riboregulator described here is not novel as similar RNA-based regulators have been reported before. It is not immediately clear if the riboregulator reported here has any advantage compared to those previously reported. In addition, the biological containment system designed to counter horizontal gene transfer was not sufficiently proven to be effective in-vivo as no gene transfer was observed even in the control strain in this study. I suggest that the following comments be addressed during revision.

Major comments

1. The engineered riboregulator is very similar to those described in previous publications (see Isaacs, F., Dwyer, D., Ding, C. et al. *Nat Biotechnol* 22, 841–847 (2004)). Please discuss and highlight the differences and advantages of the riboregulator described in this study compared to those already reported.

- We appreciate the reviewer's comments and suggestions to help us improve our manuscript. We have added detailed explanations regarding the differences and advantages of the engineered riboregulatory (ER) described in this study in **Lines 159 and 509**.

In previous work reported by Isaacs et al., crRNAs were successfully constructed in *E. coli* so that the purine-rich RBS region upstream of the start codon could be blocked by formation of the secondary structures. Unlike *E. coli*, in which the homology of the Shine-Dalgarno sequence to the 16S rRNA largely dictates the strength of translation, the strength of gene expression in *Bacteroides* correlates poorly with the level of RBS complementarity to the 16S rRNA. The RBS of *Bacteroides* tends to be AT-rich and more sensitive to secondary structure than that of *E. coli*. The putative S1 protein binding site in the RBS was reported to have a strong effect on the strength of the RBS. Therefore, we originally targeted this region to make crRNA structures and have demonstrated, for the first time, that this methodology successfully yielded an ER in *Bacteroides* spp. The same methodology could be applied to the other human commensals that, like *Bacteroides*, rely on the strength of the S1 protein binding site for translation. In addition, in our study, the ER was stably introduced into the *Bacteroides* chromosome whereas the previous work showed the function of the ER with plasmids. Genes on a chromosome are more stable than those on a plasmid in terms of HGT.

2. The temperature is likely to influence the activity of the riboregulator as the structure of the folded trans- and cis- RNA elements will change. Please discuss how this system behaves at temperatures other than physiological temperature.

- We appreciate the reviewer's comments and suggestions. We have added the discussion on the behavior of the ER system at temperatures other than physiological temperature in **Lines 196** and **524**.

Our first goal was to make sure that the ER system can work well at physiological temperature (i.e., 37°C). Moreover, we calculated the minimum free energy of the secondary structure of taRNA6 by NUPACK at temperatures ranging from 22 to 41°C. As shown in the figure below, the minimum free energy (MFE) of taRNA6 increased as temperature increased, indicating its instability in high temperatures. Temperature higher than the physiological one is likely to increase the outputs of this ER system. When the transformed bacteria come out of the body, the production of the protein would not be increased at ambient temperature, because of the increased stability of the taRNAs.

3. Fig 3f shows that the presence of the oxygen inhibits thymineless death of the bacteria due to the decrease in cell growth. Please check if other stressful conditions, such as limited nutrient availability, are likely to have the same effect on the auxotrophy in the bacteria.

- We appreciate the reviewer's comments and suggestions. We have done additional experiments to measure the auxotrophy of engineered *B. thetaiotaomicron* in several important stress conditions (**Supplementary Fig. 5; Lines 261**).

To test the effect of carbon sources, we measured the growth curve and viability of the engineered *B. thetaiotaomicron* in the presence of several carbon sources: monosaccharide (i.e., glucose and galactose), disaccharide (i.e., maltose), and polysaccharide (i.e., starch), with or without thymidine. As shown in **Supplementary Fig. 5a and b**, engineered *B. thetaiotaomicron* grew rapidly in the presence of thymidine with either a monosaccharide or a disaccharide as the carbon source. On the contrary, no viable cells were detected in the absence of thymidine after 48 hours of incubation (**Supplementary Fig. 5b**). The growth of the strain was much slower when starch was used as the sole carbon source with thymidine than when monosaccharide or disaccharide was used (**Supplementary Fig. 5b**). Significantly lower numbers of viable cells were detected, compared to those with thymidine, after 96 hours of incubation in the absence of thymidine with starch as the sole carbon source (**Supplementary Fig. 5b**).

We also tested the effect of carbon source starvation on auxotrophy in the engineered strain of *B. thetaiotaomicron* (**Supplementary Fig. 5c**). Bacterial cells were cultured in a defined minimal medium (DMM) without glucose after PBS wash in the presence or absence of thymidine. As expected, they stopped growing without a carbon source. Moreover, the viability of strains in the absence of thymidine significantly decreased after 48 and 96 hours of incubation, indicating the thymidine auxotrophy under conditions of carbon source starvation.

Considering the important roles and ubiquitous presence of bile acids in the human gut, we also tested the effect of crude bile acid and secondary bile acid (i.e., deoxycholic acid, or DCA) on the autotrophy of engineered *B. thetaiotaomicron* (**Supplementary Fig. 5d, e**). Bacteria stopped growing without thymidine (**Supplementary Fig. 5d**) and no viable cells were detected after 48 hours

incubation in the absence of thymidine in TYG medium with either crude bile acid or DCA (**Supplementary Fig. 5e**).

In summary, as major factors of stress conditions in the gut, carbon sources and bile acids could delay thymineless death; nevertheless, the engineered *B. thetaiotaomicron* retained auxotrophy, indicating the robustness and efficiency of our biocontainment.

Supplementary Fig. 5

4. In Fig 6, the evolutionary stability of ER and CD should be checked in conditions that better represent the physiological conditions to accurately predict how these genetic elements will behave once the bacteria colonize the gastrointestinal tract.

➤ We appreciate the reviewer's comments and suggestions. We have checked the evolutionary stability of ER and CD at day10 *in vivo*, as shown in **Supplementary**

Fig. 11, to evaluate the genetic stability of the ER and CD in the gastrointestinal tract. The result is described in Line 465.

We isolated four colonies of the $\Delta thyA$ (ER⁺/CD⁺) strain from the feces of mice at day 10 and compared the functions of the isolates between before and after inoculation. There was no significant difference in the luminescence values among the isolates, including controls prior to inoculation (**Supplementary Fig. 11a**). When the isolates were conjugated with *E. coli* carrying plasmids having the *thyA* gene, no viable cells were detected among the isolates derived from the feces. Thus, the CD remarkably inhibited the HGT of *thyA* gene, as shown in **Supplementary Fig. 11b**. These results reveal that the ER and CD were stable for at least 10 days in the gastrointestinal tract.

Supplementary Fig. 11

5. The in-vivo study did not prove that CD prevents the escape of the bacteria from biological containment as no horizontal gene transfer was observed in the study (as depicted by the low frequency of the escape of the strain with the non-targeted sgRNA). As the main crux of this study is the use of CD for biological containment, it should be demonstrated in an in-vivo model. Perhaps the authors can make use of a bacterial strain with a plasmid that is more readily able to undergo horizontal gene transfer into the *B. thetaiotaomicron* with ER and CD.

- We thank the reviewer for these comments and suggestions and have now included an *in vivo* study to validate that the CD functions to prevent the disruption of thymidine auxotrophy by HGT of *thyA* gene *in vivo*, as shown in **Fig.7d and e**; Line 474 and Line 580.

Germ-free (GF) mice were first colonized with *E. coli* containing a mobilizable plasmid bearing the *thyA* gene. Subsequently, to assess the plasmid-borne transfer of the *thyA* gene from *E. coli* to *B. thetaiotaomicron* in the mouse gut, thymidine-auxotrophic *B. thetaiotaomicron* strains with the ER and CD: $\Delta thyA$ (ER⁺/CD⁺), or with the ER and non-targeting CD: $\Delta thyA$ (ER⁺/CD^{NT}) were administered (**Fig. 7d**). As expected, we observed colonies of the latter strain [$\Delta thyA$ (ER⁺/CD^{NT})] on TYG agar plates both with and without thymidine when the stools of the mice collected on day 7 were plated (**Fig. 7e**). In contrast, no colony of the former strain [$\Delta thyA$ (ER⁺/CD⁺)] was detected on the TYG agar plates

without thymidine, showing that the thymidine auxotrophy persisted. This result revealed that the CD successfully prevented disruption of the thymidine auxotrophy *in vivo*. However, the preventive effect was lost by day 14 (11 days after administration of *Bacteroides* strains) as the thymidine-auxotrophic strain [$\Delta thyA$ (ER⁺/CD⁺)] was replaced by a *B. thetaiotaomicron* strain without thymidine auxotrophy. We isolated twelve colonies of the non-auxotrophic strain and performed whole genome sequencing to clarify the mechanism of escape of the thymidine auxotrophy. The sequencing analysis showed that all twelve isolates lacked transgenes and had the *thyA* gene (**BioSample accession number: SAMN37130011-SAMN37130022**), whereas the ancestral strain contained the transgenes without the *thyA* gene (**BioSample accession number: SAMN37130023**). We assume that a double crossover had occurred between the regions flanking the transgenes on the genome and homologous regions flanking the *thyA* gene on the plasmid. The transconjugant likely appeared via conjugation between *E. coli* and *B. thetaiotaomicron* after day 7, and the transconjugant with *thyA* apparently grew and outcompeted the thymidine-auxotrophic strain [$\Delta thyA$ (ER⁺/CD⁺)]. It is noteworthy that the transgenes we introduced into the wild-type *B. thetaiotaomicron* were removed upon loss of the CD and acquisition of *thyA*. Therefore, genetically engineered *B. thetaiotaomicron* with transgenes was still biocontained even after the CD failed to target the *thyA* gene, as the strain reverted back to the wild-type genotype in the *thyA* locus.

The CD successfully prevented disruption of thymidine auxotrophy for four days. However, the extended experiment showed the CD was not flawless and there is a possibility that the CD failed to guard thymidine auxotrophy beyond this time period. Improvements, for example, by adjusting the strength of promoters, or using Cas3 and the CasABCDE complex for the CD, are likely to produce a more stringent auxotrophic system. Nevertheless, this containment system was highly reliable since the genetically engineered strains with both transgenes and the *thyA* gene were not detected even if the thymidine auxotrophy was no longer functional.

Fig. 7d and e

Minor comments

1. Please provide a reference for lines 154-157 on page 7.

- We thank the reviewer for this suggestion, and we have added two references to **Line 164** (former **Line 157**): *PLoS One*. 2011;6(8):e22914 and *Appl Environ Microbiol*. 2013 Mar; 79(6): 1980–1989.

2. Figure 3b, please mention the amount of thymidine supplemented in the media.

- We thank the reviewer for this suggestion, and we have added the concentration of thymidine, which was 413 μ M (**Line 1002-1005**).

3. For the filter mating experiment in Fig 4b and 5b, the selection media was BHIS with erythromycin. Please check if this is correct as this media will not select against the *E. coli* donor.

- We thank the reviewer for this comment and have checked that this is correct. The BHIS agars contain gentamicin and thymidine in addition to erythromycin, as described in Methods (**Line 778 and 803**). To clarify that by using the antibiotics and thymidine we selected not the *E. coli* donor but, rather, transconjugants having the transferred plasmid bearing the erythromycin resistance gene, we revised the information on the components of the agar plates in the Legends of **Fig. 4** and **Fig. 5**. In fact, we have confirmed that no *E. coli* donor cells were selected by plating, as negative controls, on BHIS agar containing gentamicin, thymidine, and erythromycin.

4. Fig 7a should be redrawn. The figure currently suggests that in all groups, the WT strain was administered on Day 0.

- We thank the reviewer for this comment and have redrawn **Fig. 7a** appropriately.

5. As significant toxicity is observed due to CD, perhaps the authors can include in the discussion how this can be mitigated by using repressible/inducible systems.

- We appreciate the reviewer's suggestion. Significant toxicity of the CD was observed when the transcription of sgRNA was upregulated. On the other hand, the efficacy of the CD seemed to be affected by the amount of transcription of sgRNA. Thus, there was a tradeoff between toxicity and efficacy.

We think that disruption of the biological containment system could best be prevented by keeping the gene expression of SpCas9 and sgRNA constant at a proper level. To address this challenge, we have chosen constitutive promoters as a simpler strategy than repressible/inducible systems, and then adjusted the level of expression of sgRNA. As shown in **Fig. 3a**, **Fig. 4b** and **Fig. 5b**, this strategy mitigated the toxicity and demonstrated the function of the CD. Moreover, the CD seemed to help increase the death rate, which might be derived from its toxicity, shown in **Supplementary Fig. 4**.

6. Please discuss the likelihood and the consequence of the horizontal gene transfer of the partial gene circuit, such as only the gene of interest.

- We appreciate the reviewer's suggestion. We described in **Line 130** that the HGT of a partial gene circuit could occur by the genetic transduction of phages, and if that were to occur, the expression of the gene circuit would be suppressed by the cis-repressive sequence.

7. The authors have shown that CD fails to prevent horizontal gene transfer in case the target gene is mutated. As such, please discuss how the CD will be able to prevent the escape from biological containment as there are potentially numerous variants of the target gene in the environment.

- We appreciate the reviewer's comment and suggestion. We added the discussion in **Line 555** about how the CD will prevent escape from biocontainment when the target genes have mutations.

A preferable candidate as a target sequence would be one that was chosen from well-conserved sequences in a gene, such as *thyA* or *dapA*, whose deficiency causes auxotrophy, leading to rapid death unless an essential nutrient is supplied. Furthermore, introducing multiple sgRNAs helps cover the wider range of potential variants.

Reviewer #2 (Remarks to the Author):

The manuscript by Hayashi et al is a well written and well presented study on the engineering of a biocontainment system for engineered DNA in a *Bacteroides* strain capable of colonising the human gut microbiome. Previously this strain was put forward by this research group and others as a promising chassis organism for engineering in the human gut microbiome. A challenge for putting engineered bacteria in the human gut is biocontainment and a genetic engineering solution to that is proposed and tested here. The proposed approach centres around thymidineless death and thymidine auxotrophy - a long established method to have bacteria only able to survive when thymidine is provided. The authors here upgrade this simple system with some clever tricks designed to prevent HGT and some aspects of mutation finding a way around the trophic containment. Simply put, they surround their engineered 'cargo' gene with a CRISPR device that cuts the *thyA* gene and flank that device with a pair of optimised riboregulator parts that control the expression of the cargo gene. If the CRISPR device is deleted, then its very likely that the riboregulator parts will be separated and so cargo gene expression will not work. The whole construct is added to the genome to replace the *thyA* gene. The authors run multiple tests of different designs, especially the riboregulator parts and guide RNA choices, and show the containment system working well in vitro and in vivo too in mice models. They only ever show it's performance with a single cargo gene (NanoLuc) but this is good enough as a representation. By the end of the manuscript the system is shown to be performing very well, but falls slightly below the goal set by the NIH. Anaerobic vs. Aerobic conditions, and the presence of small amounts of thymidine naturally in the gut seem to pose problems that stem not from the engineering done here,

but in the thymidine trophic selection system in general. In the discussion the authors cover how improvements to their work could solve these minor issues.

I liked this work, and I think it is well described, innovative and timely. It is a good fit for the journal readership and I'm supportive of its publication without any major revisions needed.

➤ We appreciate the reviewer's comments.

In fact, I could only find 3 minor points that I think need addressing:

1. Line 65 - xenobiology is put forward as a biocontainment system, but the authors only discuss XNA xenobiology which I think has never been practically applied in bacteria in application. I was surprised that genetic code 'xenobiology' was not mentioned here, where non-canonical amino acids (ncAAs) are used for biocontainment. George Church lab and Farren Isaacs lab both published remarkably good biocontainment results from using the recoded E.coli genome and having TAG codons placed in essential genes in a way which requires ncAAs to be present for those genes to fold and function. The best of these is citation #13 in the manuscript, and is cited in the auxotrophy section of the intro instead. I think this is a bit of a mix-up and I would instead consider this work fitting in better as part of the 'xenobiology' section. In fact, I would probably change the name to 'genetic firewall' rather than xenobiology and then the codon recoding that prevents HGT (e.g. by phages) in these organisms can also be covered - e.g. with Jason Chin's Syn61 E.coli.

➤ We appreciate the reviewer's comments and suggestions. We have replaced the description of XNA xenobiology with that of genetic code xenobiology as genetic firewalls in **Line 66**.

Genetic firewalls use alternative genetic codes in bacteria for nonstandard amino acids (NSAAs), enabling genetic isolation by preventing horizontal gene transfer (HGT). The growth of the bacteria harboring the genetic firewall thus depends on the use of the NSAAs, such that the firewall sets up robust and orthogonal barriers between the engineered organisms and the environment. The artificial genetic code, incorporating NSAAs, makes the standard amino acids from natural microorganisms nonfunctional in the engineered bacteria. Additionally, as bacteriophage infection can lead to the HGT of bacterial genes, a genetic firewall has been created by the deletion of cellular transfer RNAs and release factor 1.

2. Figure 2h legend - please say in the legend what the green and black bars represent.

➤ We appreciate the reviewer's suggestion and have added an explanation of the green and black bars in **Line 982**.

Black (Grey) columns represent original promoters (P_1 / P_{BT1311}) whereas green ones show switched promoters (P_{BT1311} / P_1).

3. Line 335 - the promoter used presumably has to operate in a broad range of host cells for the system to work. Can the authors detail their approach to choosing a promoter that is predicted to function in a broad range of gut bacteria cells? Either here or in the discussion section.

- We appreciate the reviewer's suggestion and discussed in **Line 570** our approach to choosing a promoter that would function in a broad range of gut bacterial cells.

Transgenes are unlikely to be expressed by species not closely related to *Bacteroides* because of the specific regulation of gene expression, even if the transgenes have transferred. In choosing promoters, we considered both intraspecies HGT and HGT between closely related species. We chose the active promoters in interspecies populations of *Bacteroides* for the CD: P_{cepA} or P_{cfxA} for transcription of sgRNA, and P_{cflA} for that of SpCas9. If promoters that would function in a broader range of gut bacterial cells are needed to effectively prevent the dissemination of transgenes, genomic analysis of gut microbiota could provide information on widely distributed promoters; in fact, some promoters have already been reported to function in a variety of gut bacterial cells.

Reviewers' Comments:

Reviewer #1:

Remarks to the Author:

The authors have addressed most of my previous comments. I have a few minor comments to be addressed:

1. The authors suggest that the thymidine-auxotrophic *B. thetaiotamicron* strain acquired the *thyA* gene through double crossover recombination with the *thyA* gene on the plasmid from *E. coli*. I would like to request clarification from the authors on whether the region flanking *thyA* on the plasmid has any sequence similarity to the region flanking the transgenes in the *B. thetaiotamicron* genome. According to Figure 1, it seems that the entire *thyA* gene from the genome was replaced.
2. Figure 7e: Please include a discussion on the possibility of preventing the acquisition of the *thyA* gene via horizontal gene transfer by inserting the transgenes at a different location in the genome.
3. Line 321: Please confirm if the comparison is with conjugation in the absence of thymidine (Figure 4b) rather than in the presence of thymidine.

Reviewer #2:

Remarks to the Author:

It's nice to see this article return after such a long time. The authors replies to my comments are all fine, and so I'm happy to recommend this to be accepted.

Cas9-Assisted Biological Containment of a Genetically Engineered Human Commensal Bacterium and Genetic Elements

NCOMMS-21-44334A

Point-By-Point Response to Reviewers' Comments

We thank both reviewers and the editor for their time reviewing our manuscript. We have responded to their concerns and described the changes we have made in response to each comment. In addition, we have corrected **Supplementary Fig 3a** because crRNA (crRCN)/taRNA4 complex was plotted accidentally with incorrect value of bioluminescent signal. This correction does not affect the original description in **Line 192**. We appreciate both reviewers' inputs in helping us achieve this improved version of our manuscript.

Reviewer #1 (Remarks to the Author):

The authors have addressed most of my previous comments. I have a few minor comments to be addressed:

1. The authors suggest that the thymidine-auxotrophic *B. thetaiotaomicron* strain acquired the *thyA* gene through double crossover recombination with the *thyA* gene on the plasmid from *E. coli*. I would like to request clarification from the authors on whether the region flanking *thyA* on the plasmid has any sequence similarity to the region flanking the transgenes in the *B. thetaiotaomicron* genome. According to Figure 1, it seems that the entire *thyA* gene from the genome was replaced.

- We appreciate the reviewer's comment and suggestion to help us improve our manuscript. As shown in **Supplementary Fig. 2a**, the entire *thyA* gene was replaced by double crossover at the upstream (5'-*thyA*, approximately 1000bp) and downstream (*thyA*-3', approximately 1000bp) of *thyA* gene. Schematic of the plasmid construct, used in the *in vivo* assay for inhibition of horizontal gene transfer of *thyA* gene, is illustrated in **Fig. 4a**. The plasmid has the sequence from 5'-*thyA* through *thyA*-3' including *thyA* gene. Therefore, both of the genome of *B. thetaiotaomicron* and the plasmid possess the identical sequences of 5'-*thyA* and *thyA*-3'. To clarify that the sequence similarity, we added the explanatory description below in **Line 495**.

Both of the genome and the plasmid have the homologous sequences at the upstream (5'-*thyA*) and downstream (*thyA*-3') of the *thyA* gene, as shown in **Supplementary Fig. 2a** and **Fig. 4a**.

2. Figure 7e: Please include a discussion on the possibility of preventing the acquisition of the *thyA* gene via horizontal gene transfer by inserting the transgenes at a different location in the genome.

- We appreciate the reviewer's suggestion to help us improve our manuscript. We have added a discussion on the possibility of preventing the acquisition of the *thyA* gene by inserting the transgenes at the other site of *thyA* locus on the genome of *B. thetaiotaomicron* in **Line 586**.

The failure of the CD could be prevented by introducing transgenes at the other sites of *thyA* locus in a *thyA*-deficient strain. This genetic modification avoids loss of the CD which would be caused by genetic exchange between the transgenes and an exogenous *thyA* gene.

3. Line 321: Please confirm if the comparison is with conjugation in the absence of thymidine (Figure 4b) rather than in the presence of thymidine.

- We appreciate the reviewer's critical comment. The comparison in **Line 321** is with conjugation in the absence of thymidine. We revised the incorrect word.

Reviewer #2 (Remarks to the Author):

It's nice to see this article return after such a long time. The authors replies to my comments are all fine, and so I'm happy to recommend this to be accepted.

- We appreciate the reviewer's kind comment.